# Sweet neurons inhibit texture discrimination by signaling TMC-expressing mechanosensitive neurons in *Drosophila*

**Shun-Fan Wu[1,2]\*, Ya-Long Ja[1], Yi-jie Zhang[1], Chung-Hui Yang[2]\***

[1]State & Local Joint Engineering Research Center of Green Pesticide Invention and Application, College of Plant Protection, Nanjing Agricultural University, Nanjing, China; [2]Department of Neurobiology, Duke University, Durham, United States

**Abstract** Integration of stimuli of different modalities is an important but incompletely understood process during decision making. Here, we show that *Drosophila* are capable of integrating mechanosensory and chemosensory information of choice options when deciding where to deposit their eggs. Specifically, females switch from preferring the softer option for egg-laying when both options are sugar free to being indifferent between them when both contain sucrose. Such sucrose-induced indifference between options of different hardness requires functional sweet neurons, and, curiously, the Transmembrane Channel-like (TMC)-expressing mechanosensitive neurons that have been previously shown to promote discrimination of substrate hardness during feeding. Further, axons of sweet neurons directly contact axons of TMC-expressing neurons in the brain and stimulation of sweet neurons increases $Ca^{2+}$ influx into axons of TMC-expressing neurons. These results uncover one mechanism by which *Drosophila* integrate taste and tactile information when deciding where to deposit their eggs and reveal that TMC-expressing neurons play opposing roles in hardness discrimination in two different decisions.
DOI: https://doi.org/10.7554/eLife.46165.001

\*For correspondence:
wusf@njau.edu.cn (S-FW);
rebecca.yang@gmail.com (C-HY)

**Competing interests:** The authors declare that no competing interests exist.

## Introduction

In addition to detecting and discriminating stimuli of different modalities, animals frequently need to integrate information detected by different sensory systems (*Stein et al., 1988*; *Sánchez-Alcañiz and Benton, 2017a*). The demand for sensory integration is particularly pressing when animals must choose among options that differ along several independent dimensions (*McFarland, 1977*). For example, when assessing food options that differ in appearance, taste, smell, and texture, it may be advantageous for animals to integrate different sensory signals associated with each option and convert them into values along the same scale so that these options can be directly compared. While significant progress has been made in identifying the molecules and neurons that sense and process different sensory stimuli (e.g. *Dunipace et al., 2001*; *Wang et al., 2004*; *Dahanukar et al., 2007*; *Cameron et al., 2010*; *Weiss et al., 2011*; *Fujii et al., 2015*; *Zhang et al., 2016*; *Sánchez-Alcañiz et al., 2017b*), less is known about how animals integrate different sensory information during decision making.

Egg-laying site selection by *Drosophila* females is a promising model for studying the molecular and circuit basis of sensory integration during decision making (*Yang et al., 2008*; *Joseph et al., 2009*). *Drosophila* females lay one egg at a time and will explore and assess available substrates prior to depositing each egg (*Yang et al., 2008*). Further, previous studies have shown that *Drosophila* females are sensitive to different properties of potential substrates such as sweetness

(*Yang et al., 2015*; *Schwartz et al., 2012*), hardness (*Karageorgi et al., 2017*), and levels of illumination (*Guntur et al., 2015*; *Zhu et al., 2014*; *Guntur et al., 2017*), to name a few. Further, *Drosophila* females will reject an 'inferior' but otherwise acceptable substrate if a better one is available, thus hinting that they might be capable of ranking their options by integrating different sensory stimuli associated with each option and converting them into signals that reflect values (*Yang et al., 2008*; *Joseph et al., 2009*; *Schwartz et al., 2012*; *Yang et al., 2015*; *Joseph and Heberlein, 2012*; *Azanchi et al., 2013*). While the specific circuit mechanisms by which *Drosophila* females integrate different sensory signals during egg-laying remain to be determined, it is a tractable problem given the relative ease in assaying egg-laying preferences of many individual *Drosophila* females in parallel and the large number of gene and circuit manipulation tools available for this model organism (*Venken et al., 2011*).

In this work, we set out to assess how *Drosophila* females integrate information of two distinct sensory properties of potential substrates – sweetness and hardness – when deciding where to deposit their eggs. Previous reports have shown that flies generally prefer the softer of two sweet substrates during feeding; for example, they prefer 0.5% agarose with sugar over 2.0% agarose with sugar (*Jeong et al., 2016*; *Zhang et al., 2016*; *Sánchez-Alcañiz et al., 2017b*). This feeding preference requires inputs from at least two distinct groups of mechanosensitive neurons on the labellum: one that uses the evolutionarily conserved mechanosensitive channel Transmembrane Channel-like (TMC) to sense hardness of substrates during discrimination of substrates, whereas the other uses the TRP channels NompC and Nanchung (Nan) (*Jeong et al., 2016*; *Zhang et al., 2016*; *Sánchez-Alcañiz et al., 2017b*). Interestingly, activation of the group that expressed NompC and Nan channels can directly inhibit the output of the sweet-sensing taste neurons, thereby further increasing the 'food value' of the softer substrate – the harder substrate should taste less sweet than the softer one because it induces stronger activation of the NompC/Nan-expressing neurons (that then exerts stronger suppression of output of sweet neurons) (*Jeong et al., 2016*). Inspired by these earlier reports, here we probed how *Drosophila* discriminate substrates of different hardness when making egg-laying decisions and how such decision is modified by detection of sweetness on the substrates. We found that, as in feeding, flies also generally preferred the softer substrate when given a hard substrate and a soft one to choose from for egg-laying. However, their soft preference during egg-laying did not require input from either of the two groups of mechanosensitive neurons that are critical for sensing and discriminating hardness of food substrates during feeding. More interestingly, detection of sweetness on substrates reduced discrimination of egg-laying substrates of different hardness, likely due to direct enhancement of output of TMC-expressing neurons by sweet neurons via axon-axon communication. Our results thus uncover a new mechanism by which *Drosophila* integrate chemosensory and mechanosensory information on choice options during an important decision-making task and highlight the fact that flies can recruit different sensory mechanisms to integrate the same two stimuli depending on the nature of their decision tasks – that is whether they are choosing substrates for feeding or for egg-laying purposes.

## Results

### *Drosophila* are receptive to laying eggs on plain (sugar free) substrates of different hardness but generally prefer the softer one

While it is commonly observed that flies tend to avoid depositing their eggs on very hard surfaces (e.g. hard plastic), the specific range of hardness that is acceptable for flies to lay their eggs and how well they discriminate egg-laying substrates of different levels of hardness have not been extensively explored. Since previous reports have shown that agarose of different concentrations have different levels of hardness (*Jeong et al., 2016*; *Sánchez-Alcañiz et al., 2017b*; *Karageorgi et al., 2017*), we first presented egg-laying females with different concentrations of agarose in our high-throughput egg-laying apparatus, each of which can assay egg-laying of 30 individual females in parallel (*Gou et al., 2016*). Importantly, we did not add sucrose or other nutritive substances to these substrate so as to minimize the impact of chemosensory cues on flies' sensing and discrimination of substrate hardness. Instead, we placed a drop of sucrose solution some distance away from the agarose substrates so that flies still had access to an energy source during egg-laying (see the 'center hole' in *Figure 1A*).

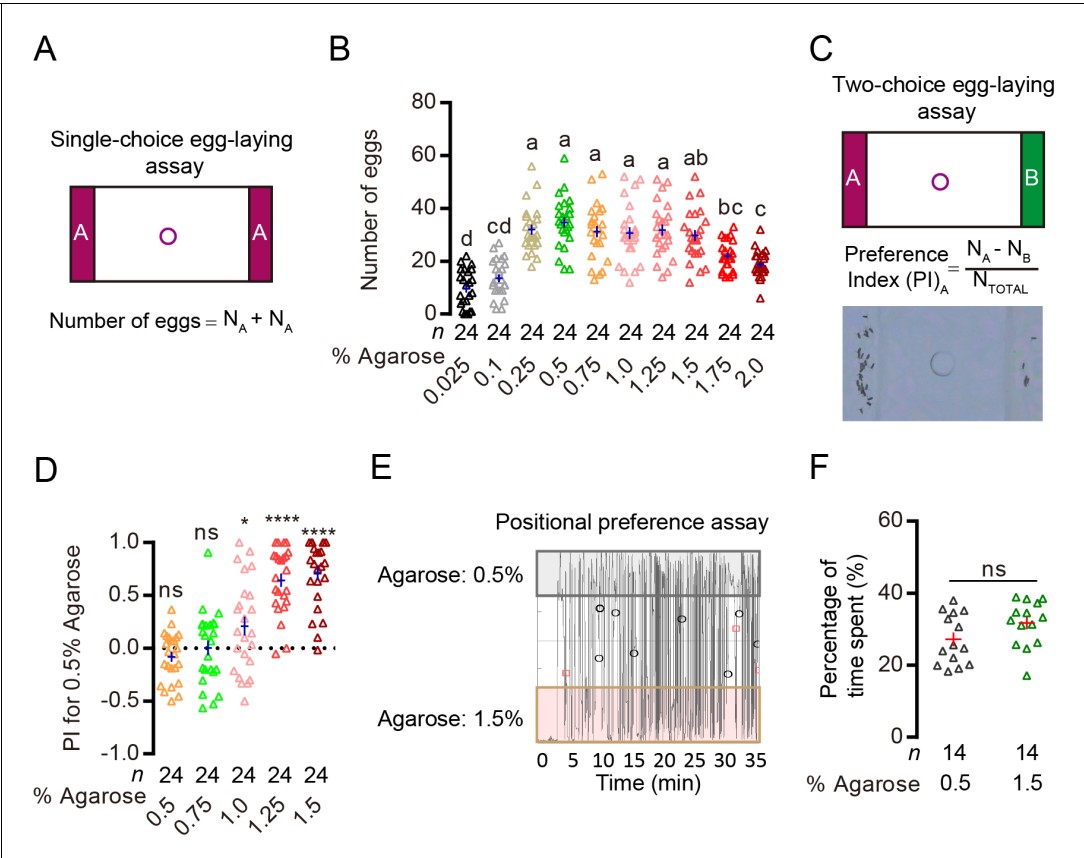

**Figure 1.** *Drosophila* can discriminate egg-laying substrates of different hardness. (**A**) Upper image: schematic of our single-choice assay. In this assay, we placed the same agarose (colored strips) on the two sides of the arena and a drop of sucrose in the center hole. The two agarose are separated by acrylic. Each of our apparatus has 30 arenas that can assay egg-laying of 30 individual females. (**B**) Comparison of acceptance of different concentrations of agarose for egg-laying in single-choice assay. Each data point in a column denotes the number of eggs laid by a single female over 14 hr. The numbers of females examined per group are labeled on the graph (N = 24 for each group in this experiment). Note that in this work, when comparisons of multiple groups were needed, we used letters (e.g. **a**, **b**) to describe the statistical relationship between them and used the following rule: groups that share at least one letter (e.g. **ab** vs. **bc**) are statistically indistinguishable, and groups that have different letters (e.g. **a** vs. **b**) are statistically different. One-way ANOVA followed by Tukey's multiple comparisons test. These comparisons may yield different p values (e.g. $p<0.05$, $p<0.0001$) at times, in which case, we labeled the highest. Also, throughout this work, the 'cross' labeled in each column denotes sample mean ± s.e.m. (**C**) Upper panel: schematic of our two-choice assay. In this assay, we placed two different agarose (colored strips) on the two sides of the arena. Lower panel: formula for calculating egg-laying preference index (PI) and a representative image of eggs laid by a single WT female in a 0.5% vs. 1.5% two-choice assay. (**D**) PI (for 0.5% agarose) of WT($w^{1118}$) females in different two-choice assays where 0.5% agarose was pitted against other concentrations of agarose. ns: not significant, ****$p<0.0001$, *$p<0.05$; Wilcoxon signed-rank test ($H_0 = 0$). (**E**) Representative trajectory of a single WT female as it explored the arena in a 0.5% vs. 1.5% two-choice assay. The x-axis denotes time, and the y-axis denotes the y position of the fly. (**F**) Quantification of the proportion of time females spent on the 0.5% agarose vs. on the 1.5% agarose in the two-choice arena. ns: not significant; Wilcoxon matched-pairs test.

DOI: https://doi.org/10.7554/eLife.46165.002

The following source data is available for figure 1:

**Source data 1.** Raw numerical data for *Figure 1*.
DOI: https://doi.org/10.7554/eLife.46165.003

To assess the acceptability of substrates of different hardness for egg-laying, we first presented flies with agarose of different concentrations in single-choice conditions where they had access to agarose of one specific concentration in each chamber (*Figure 1A*). We found that flies laid comparable numbers of eggs on agarose whose concentration ranged from 0.25% to 1.5% (*Figure 1B*). But they laid fewer eggs on agarose with concentrations that were either too high or too low (*Figure 1B*), likely because they cannot easily insert their ovipositor into a substrate that is very hard and because they tend to avoid walking on a surface that is liquid-like. We then determined how

well females discriminated agarose of different concentrations in two-choice assays (*Figure 1C*). In particular, we focused on examining whether flies favored the softer 0.5% agarose over several other harder substrates. We found that flies consistently laid more eggs on the 0.5% agarose when it was pitted against 1.0%, 1.25% or 1.5% agarose in different two-choice assays (*Figure 1D*). Since flies laid comparable amount of eggs on these concentrations of agarose in single-choice assays (*Figure 1B*), their preference for the softer 0.5% agarose in the two-choice assays indicated that: 1) they were capable of discriminating 'equally acceptable' substrates of different hardness and 2) they preferred the softer one when given a choice. Further, tracking the positions of flies as they explored and laid eggs in a 0.5% vs. 1.5% two-choice assay showed that they spent comparable amounts of time on these two agarose substrates (*Figure 1E and F*), suggesting that their preference to deposit eggs on the softer 0.5% agarose substrate was not due to intrinsic positional preference for the softer substrate. We thus chose the 0.5% vs. 1.5% agarose as our standard two-choice assay to search for the specific sensory mechanism that enables *Drosophila* females to discriminate substrates of different hardness during egg-laying.

## *Drosophila* use different mechanosensitive channels and neurons to discriminate substrates of different hardness during feeding and egg-laying

To start identifying the potential sensory mechanism by which egg-laying females discriminate agarose substrates of different hardness, we screened a small collection of mutants for genes that have been shown to sense water, hardness of food substrates during feeding, or nociceptive mechanical force. This list includes mutants for the water sensor Ppk28 (*Cameron et al., 2010*), TRP channels Nanchung (Nan) (*Kim et al., 2003*; *Gong et al., 2004*), Inactive (Iav) (*Kim et al., 2003*; *Gong et al., 2004*), NompC (*Walker et al., 2000*), and the evolutionarily conserved mechanosensitive channels Piezo (*Kim et al., 2012*) and TMC (*Zhang et al., 2016*). Curiously, none of these mutants behaved differently from controls: they all showed a preference to deposit their eggs on the softer 0.5% agarose substrate when given a 0.5% and a 1.5% agarose substrate to choose from (*Figure 2A*).

We next attempted to identify the specific sensory neurons that the flies use to discriminate substrates of different hardness during egg-laying. Previous studies have shown that *Drosophila* prefer to feed on softer substrates over hard ones, and that they rely on two distinct groups of mechanosensitive neurons on the labellum to sense and discriminate hardness of substrates (*Zhang et al., 2016*; *Jeong et al., 2016*; *Sánchez-Alcañiz et al., 2017b*). The first group has only two members that are known as the *m*ulti-*d*endritic neurons in the *l*abellum (MD-L neurons), displays large and complex dendrites, and expresses and uses the evolutionarily conserved mechanosensitive channel TMC to sense the hardness of food substrates (*Zhang et al., 2016*; *Pan et al., 2018*; *Vreugde et al., 2002*; *Keresztes et al., 2003*; *Pan et al., 2013*). The second group has numerous members, displays dendrites that are short and simple, and expresses and uses the TRP channels Nanchung (Nan) and NompC to sense the hardness of food substrates (*Jeong et al., 2016*; *Sánchez-Alcañiz et al., 2017b*). We wondered whether these two groups of mechanosensitive neurons may also play a role in discriminating substrates of different hardness during egg-laying, although possibly by using channels other than TMC, NompC, and Nan, since mutants for these three channels did not behave significantly different from controls (*Figure 2A*). However, silencing these neurons by using the same tools employed by the feeding studies (*Jeong et al., 2016*; *Zhang et al., 2016*; *Sánchez-Alcañiz et al., 2017b*) did not significantly impact flies' ability to discriminate substrates of different hardness during egg-laying either: these manipulated flies still clearly preferred to lay eggs on the 0.5% agarose in the 0.5% vs. 1.5% two-choice assays (*Figure 2B–D*). These results raised two possibilities: first, flies may rely on an as-yet-unidentified group (or groups) of mechanosensitive neurons on the labellum to sense and discriminate substrates of different hardness during egg-laying; second, the mechanosensitive neurons that are critical for discriminating substrate hardness during egg-laying may not be housed on the labellum.

To attempt to identify the appendage(s) that houses the specific mechanosensitive neurons that are critical for discriminating substrate hardness during egg-laying, we surgically removed different appendages from the animals. While rather crude, this approach allowed us to directly assess the requirement of specific appendage in hardness discrimination during egg-laying. We severed different appendages one at a time and found that, first, not surprisingly, removing the ovipositor caused animals unable to lay any eggs (*Figure 2—figure supplement 1A*). Thus, we were unable to assess

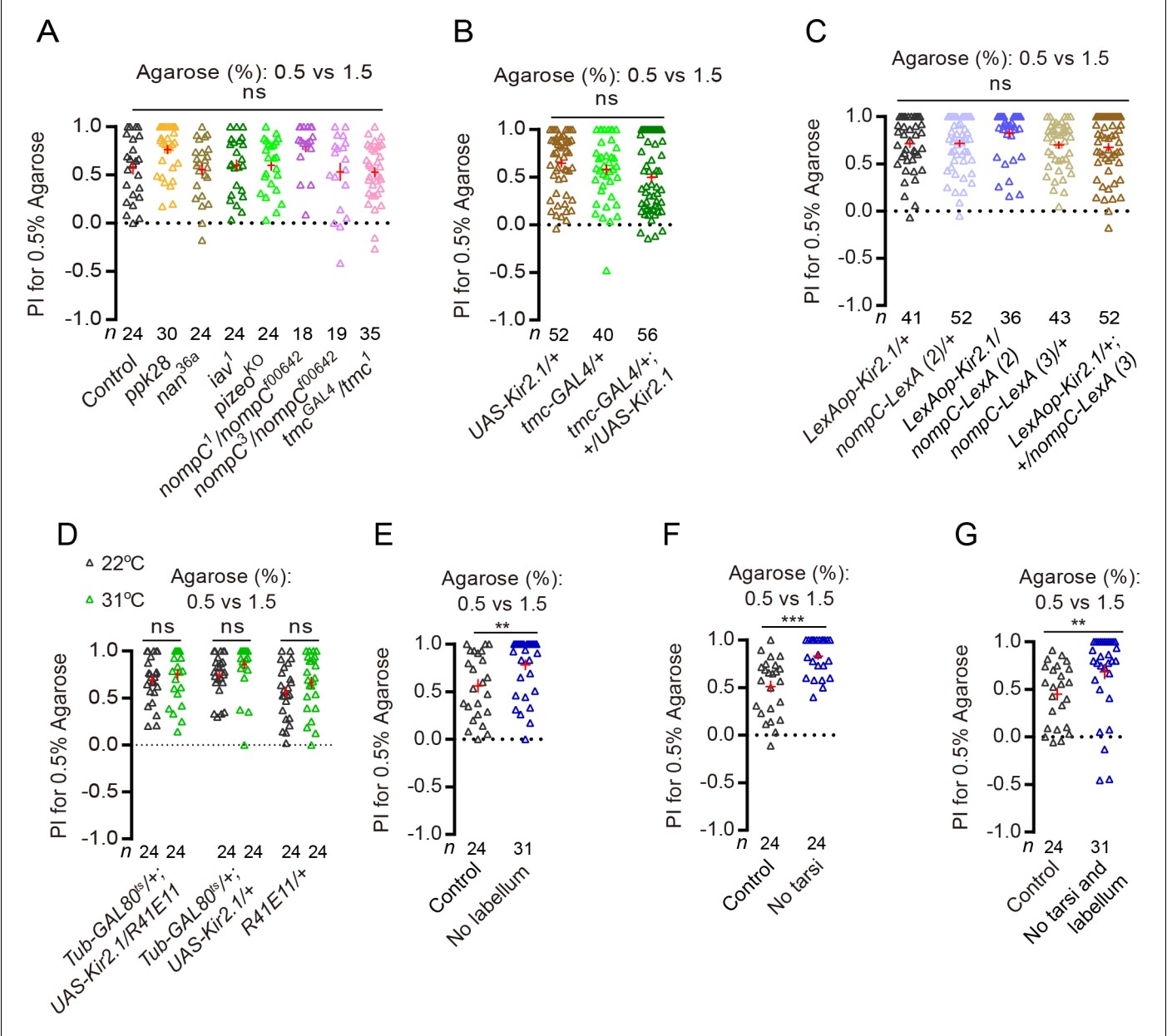

**Figure 2.** *Drosophila* use different channels and neurons to discriminate substrate hardness during feeding and egg-laying. (**A**) PI (for 0.5% agarose) of several known channel mutants in a 0.5% vs. 1.5% two-choice assay. ns: not significant; Kruskal–Wallis test followed by Dunn's multiple comparisons test. (**B–D**) PI (for 0.5% agarose) of females with TMC-expressing neurons silenced (**B**) or NompC and Nan-expressing neurons silenced (**C–D**) in a 0.5% vs. 1.5% two-choice assay. (*R41E11-GAL4* is another driver that labels the NompC and Nan-expressing neurons) (*Jeong et al., 2016*). ns: not significant; Kruskal–Wallis test followed by Dunn's multiple comparisons test for comparisons in panels (**I and J**), Mann–Whitney test for comparison in panel **K**). (**E–G**) PI (for 0.5% agarose) of WT females with their (**E**) labellum, (**F**) tarsi of all six legs, and (**G**) labellum plus tarsi of all legs removed in a 0.5% vs. 1.5% two-choice assay. ***p<0.001, **p<0.01; Mann–Whitney test.

DOI: https://doi.org/10.7554/eLife.46165.004

The following source data and figure supplements are available for figure 2:

**Source data 1.** Raw numerical data for *Figure 2*.
DOI: https://doi.org/10.7554/eLife.46165.007

**Figure supplement 1.** Contribution of different appendages on egg-laying rate and discrimination of substrate hardness.
DOI: https://doi.org/10.7554/eLife.46165.005

**Figure supplement 1—source data 1.** Raw numerical data for *Figure 2—figure supplement 1*.
DOI: https://doi.org/10.7554/eLife.46165.006

the role of ovipositor in hardness discrimination during egg-laying. However interestingly, we found that severing the rest of the appendages either singly or together did not cause any reduction in substrate hardness discrimination during egg-laying (*Figure 2E–G*; *Figure 2—figure supplement 1B–D*). More strikingly, flies without labellum or tarsi from all six legs – or both – not only still showed a clear preference to lay eggs on the softer of two substrates, their preference for the softer substrate was significantly stronger than controls (*Figure 2E–G*), suggesting that these appendages may house mechanosensitive neurons that can act to *inhibit* hardness discrimination during egg-laying.

Taken together, these results uncover three important differences between substrate hardness discrimination during feeding and egg-laying. First, flies use different sets of mechanosensitive channels to discriminate substrates of different hardness during feeding and egg-laying – TMC, NompC, and Nan were not required for discrimination during egg-laying (*Figure 2A*) but were required for discrimination during feeding (*Jeong et al., 2016*; *Zhang et al., 2016*; *Sánchez-Alcañiz et al., 2017b*). Second, although flies rely on mechanosensitive neurons on the labellum/proboscis to discriminate substrates of different hardness during feeding (*Jeong et al., 2016*; *Zhang et al., 2016*; *Sánchez-Alcañiz et al., 2017b*), the principal mechanosensitive neurons they use to discriminate hardness during egg-laying likely may be housed on the ovipositor instead since removal of none of the other appendages caused females unable to discriminate substrates of different hardness (*Figure 2G*). While we do not have direct evidence to support this claim, we note that ovipositor is known to possess mechanosensitive neurons (*Sánchez-Alcañiz and Benton, 2017a*; *Stocker, 1994*; *Newland and Burrows, 1994*) and that flies have been shown to actively probe substrates with their ovipositor prior to depositing each egg (*Yang et al., 2008*). Third and most curiously, labellum and legs may house mechanosensitive neurons whose input acts to inhibit – as opposed to promote – discrimination of substrates of different hardness during egg-laying, as severing these appendages enhanced discrimination of substrates of different hardness (*Figure 2E–G*).

## Detection of sucrose on substrates inhibits discrimination of substrates of different hardness during egg-laying

Our experiments have so far focused on assessing how egg-laying females discriminated plain agarose substrates of different levels of hardness. To begin to probe how they evaluate substrates of different hardness in the presence of additional sensory cues, we next asked whether adding sugars to substrates of different hardness affects how they are discriminated by females. Specifically, we added the same amount of sucrose to both the 0.5% and the 1.5% agarose substrates and found that, interestingly, presence of sucrose in both substrates significantly reduced females' preference to lay eggs on the softer 0.5% substrate (*Figure 3A and B*). The reduction in preference was mild when sucrose concentration was low (e.g. 5 mM) but became stronger as the concentration of sucrose increased (*Figure 3—figure supplement 1A*). When the concentration reached 100 mM, flies became virtually indifferent between the 0.5% and the 1.5% choice (*Figure 3A and B*; *Figure 3—figure supplement 1A*). A similar reduction in discrimination was observed when sucrose was replaced with several other sweet compounds, pineapple juice, but not with sorbitol, which is a nutritious but unsweet substance for flies (*Figure 3—figure supplement 1B*). In contrast, presence of the bitter chemical caffeine or the sour chemical acetic acid did not significantly impact discrimination of substrates of different hardness (*Figure 3—figure supplement 1C*). Furthermore, because a previous report has shown that arena size can influence how animals behaviorally respond to sucrose during egg-laying site selection (*Schwartz et al., 2012*), we also examined whether sucrose was similarly capable of reducing discrimination of substrates of different hardness when animals were selecting for egg-laying substrates in larger arenas. We found that, similar to what we have observed in our 'regular arenas', flies in arenas that were ~8X the size still preferred the softer 0.5% agarose over the 1.5% agarose when both substrates were sucrose free but reduced their soft preference when both contained 100 mM sucrose (*Figure 4—figure supplement 2B and C*). Together, these results showed that the presence of sweet compounds in substrates can cause females to become less discriminating between substrates of different hardness during egg-laying.

There are at least two explanations for females' reduced preference for the softer substrate when sucrose was added to substrates of different hardness. First, presence of sucrose significantly altered the hardness of these two substrates, rendering them to have similar levels of hardness. Second, detection of sucrose by sweet neurons modified how these substrates were perceived by flies. To

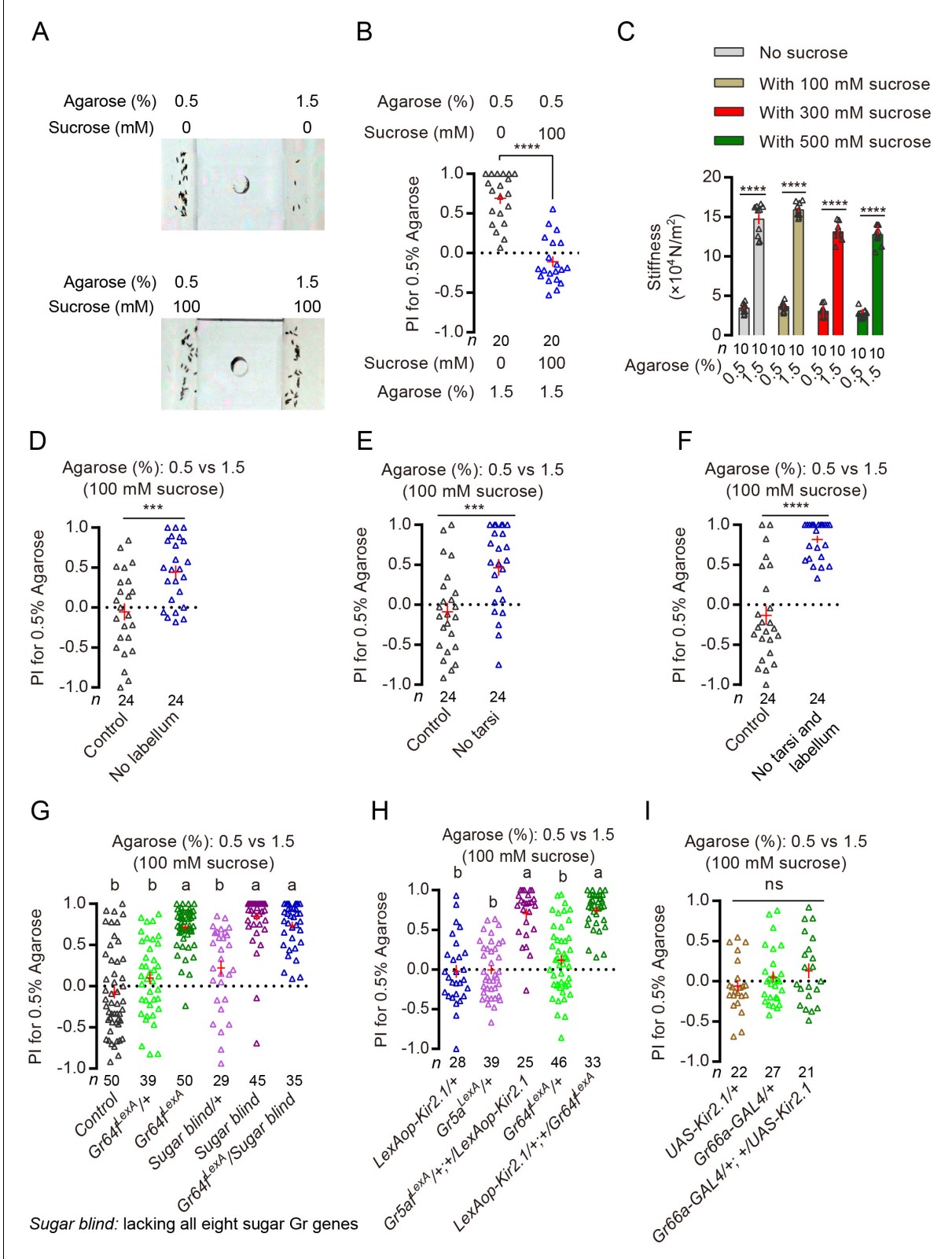

**Figure 3.** Detection of sucrose on substrates by sweet neurons can inhibit discrimination of substrates of different hardness. (A) Representative images of eggs laid by a single WT female in a 0.5% vs. 1.5% two-choice assay where both substrates were sugar free (top) and where both substrates contained 100 mM sucrose (bottom). (B) PI (for 0.5% agarose) of WT females in a 0.5% vs. 1.5% two-choice assay where both substrates were sugar free (black) and where both substrates contained 100 mM sucrose (blue). ****p<0.0001; Mann–Whitney test. The PI for the sucrose-containing group on the
*Figure 3 continued on next page*

*Figure 3 continued*

right is not significantly different from 0; Wilcoxon signed-rank test (H$_0$ = 0). (C) Stiffness of agarose 0.5% and 1.5% agarose substrates with or without 100 mM, 300 mM, and 500 mM of sucrose in them. ****p<0.0001; Mann–Whitney test. (D–F) PI (for 0.5% agarose) of WT females with different appendages surgically removed in a sucrose + 0.5% vs. sucrose + 1.5% two-choice assay. ****p<0.0001, ***p<0.001; Mann–Whitney test. (G) PI (for 0.5% agarose) of mutants that lacked either a critical co-receptor Gr64f (*Gr64f*$^{LexA}$) (*Yavuz et al., 2014*) for sugar sensing or all eight known sugar receptors ('sugar blind') (*Yavuz et al., 2014*) in a sucrose + 0.5% vs. sucrose + 1.5% two-choice assay. Groups that share at least one letter are statistically indistinguishable; Kruskal–Wallis test followed by Dunn's multiple comparisons test with p<0.05. (H) PI (for 0.5% agarose) of females with their *Gr64f*$^{LexA}$ or *Gr5a*$^{LexA}$-labeled neurons selectively silenced in a sucrose + 0.5% vs. sucrose + 1.5% two-choice assay. Groups that share at least one letter are statistically indistinguishable; Kruskal–Wallis test followed by Dunn's multiple comparisons test with p<0.001. (I) PI (for 0.5% agarose) of females with their *Gr66a-GAL4*-labeled neurons (*aka* the bitter-sensing taste neurons) selectively silenced in a sucrose + 0.5% vs. sucrose + 1.5% two-choice assay. ns: not significant; one-way ANOVA followed by Tukey's multiple comparisons test.

DOI: https://doi.org/10.7554/eLife.46165.008
The following source data and figure supplements are available for figure 3:

**Source data 1.** Raw numerical data for *Figure 3*.
DOI: https://doi.org/10.7554/eLife.46165.011
**Figure supplement 1.** Sugar sensing by sweet-taste neurons is responsible for sucrose-induced indifference between substrates of different hardness.
DOI: https://doi.org/10.7554/eLife.46165.009
**Figure supplement 1—source data 1.** Raw numerical data for *Figure 3—figure supplement 1*.
DOI: https://doi.org/10.7554/eLife.46165.010

test the first possibility, we measured the hardness of 0.5% and 1.5% agaroses with or without 100 mM, 300 mM, and 500 mM sucrose. We found that while the presence of different concentrations of sucrose did slightly alter the hardness of both the 0.5% and the 1.5% agarose substrates, differences in hardness between the two substrates remained significantly different (*Figure 3C*). To test the second possibility, we eliminated females' sweet-sensing ability and tested how well they discriminated substrates of different hardness in the presence of sucrose. We first severed different appendages that are known to house sweet neurons and found that eliminating the tarsi or labellum significantly increased flies' preference for the softer substrate in the presence of sucrose (*Figure 3D–F*). Interestingly, while severing neither the wings nor the maxillary palps had any significant effect, severing the antennae also increased soft preference in the presence of sucrose (*Figure 3—figure supplement 1D–F*). (It is worth noting that recent reports have suggested that some olfactory neurons express specific molecular sugar receptors and these neurons thus may *potentially* be capable sensing sugars, too [*Fujii et al., 2015*; *Yavuz et al., 2014*]). We next removed several molecular sugar receptors from flies and found that mutants that are known to be defective in sugar sensing (*Fujii et al., 2015*; *Yavuz et al., 2014*) also showed a clear preference for the softer substrate in the presence of sucrose (*Figure 3G*), as did flies with their sweet neurons selectively silenced (*Figure 3H*; *Figure 3—figure supplement 1G*). Further, silencing only the sweet-sensing neurons on the legs but not on the labellum (labeled by *Gr64a*$^{GAL4}$) (*Fujii et al., 2015*) recovered some soft preference (*Figure 3—figure supplement 1G*), too, albeit to a lesser extent than silencing all sweet neurons. As a control, we also silenced the bitter-sensing taste neurons and found that it failed to recover significant soft preference in the presence of sucrose (*Figure 3I*). Lastly, we showed that the improvements of hardness discrimination in the presence of sucrose by inhibiting sweet sensing were not because sweet neurons can act as inhibitors of hardness discrimination *in general*: silencing sweet neurons did not enhance discrimination of substrates of different hardness when the substrates were sugar free (*Figure 3—figure supplement 1H and I*).

Taken together, these results suggest that presence of sweet compounds on egg-laying substrates of different hardness can cause females to become less discriminating between them and reduce their preference for the softer one. Further, such sucrose-induced indifference towards substrates of different hardness requires activation of sweet neurons on flies' labellum, tarsi, and possibly also on antennae (even though it is presently unclear whether *Gr64f*-expressing neurons on the antennae are truly capable of sensing sugar).

## Sucrose-induced inhibition of discrimination of substrates of different hardness requires TMC and TMC-expressing mechanosensitive neurons

We next aimed to determine how activation of sweet neurons may promote indifference between two substrates of different hardness in the presence of sucrose. In particular, we wondered whether activation of sweet neurons may act to modulate specific mechanosensory pathways. To test this idea, we first examined the various channel mutants we had tested earlier. While none of these channels appeared to play a role in hardness discrimination when the substrates were sugar free (*Figure 2A*), it is conceivable that some may contribute to sugar-induced indifference between substrates of different hardness. Indeed, whereas most of these mutants behaved similarly to controls (*Figure 4A*), several independently generated *tmc* mutants (*Zhang et al., 2016*; *Guo et al., 2016*) behaved differently from the rest and showed a strong preference for the softer substrate despite the presence of sucrose (*Figure 4A and B*; *Figure 4—figure supplement 1A*). These results suggest TMC channel, like sweet neurons, may act to inhibit discrimination of substrates of different hardness in the presence of sucrose.

To confirm the role of TMC-dependent mechanosensation in inhibiting discrimination of substrates of different hardness in the presence of sucrose, we did the following experiments. First, we reintroduced the *tmc* gene back to the *tmc^1^* mutants (by using a *tmc-GAL4* that has been shown to label the TMC-expressing mechanosensitive MD-L neurons on the labellum as well as a few CNS neurons [*Zhang et al., 2016*]). We found that this manipulation reverted *tmc^1^* mutants to behave more like WT animals: they were less discriminating of substrates of different hardness in the presence of sucrose (*Figure 4C*). Second, we selectively silenced the output of TMC-expressing neurons (by using the same *tmc-GAL4* to direct expression of Kir2.1 or TNT [*Sweeney et al., 1995*]) and found that these manipulations caused animals to clearly prefer the soft substrate like *tmc^1^* mutants did despite the presence of sucrose (*Figure 4D* and *Figure 4—figure supplement 1B*). In contrast, silencing the Nan-expressing group of neurons on the labellum did not recover any significant soft preference (*Figure 4—figure supplement 1C*). Importantly, silencing these *tmc-GAL4*-expressing neurons in the presence of *vGlut-GAL80* (*Bussell et al., 2014*), a transgene that blocked *GAL4*-dependent expression in CNS neurons labeled by *tmc-GAL4*, still caused animals to show a clear preference for the softer substrate in the presence of sucrose (*Figure 4D–F*). But silencing neurons labeled by an independently generated *tmc^GAL4^* (a knocked-in *GAL4* for *tmc* [*Guo et al., 2016*]), which fortuitously *did not* label the TMC-expressing mechanosensitive MD-L neurons on the labellum (*Figure 4H and I*), failed to recover significant soft preference in the presence of sucrose (*Figure 4G*). We then asked whether animals that lacked either the *tmc* gene or functional TMC-expressing neurons might be defective in sugar sensing when selecting for egg-laying site, as such defect might also explain their preference for the softer substrate in the presence of sucrose. However, we found that these mutants readily discriminated two 0.5% agarose substrates with different levels of sucrose (*Figure 4—figure supplement 1D–F*), a task that requires them to sense sugar properly. We note that this is also consistent with the previous finding that MD-L neurons from WT and *tmc* mutants showed comparable electrophysiological responses to sucrose (*Zhang et al., 2016*).

Taken together, our results showed that TMC and TMC-expressing mechanosensitive neurons on the labellum did not play a significant role in hardness discrimination when both egg-laying substrates were sugar free, however interestingly, they can act to inhibit discrimination of substrates of different hardness when sucrose was present on both. The requirement of functional TMC and TMC-expressing neurons in sucrose-induced inhibition of discrimination (or sucrose-induced indifference) thus suggests that activation of sweet neurons may act to inhibit such discrimination by modulating the sensory pathway activated by TMC neurons.

## Axon termini of sweet taste neurons in the brain contact with and can signal to those of TMC neurons

Finally, we set out to identify the potential mechanism by which sweet neurons may modulate the sensory pathway activated by the TMC-expressing mechanosensitive neurons. A previous report had shown that sweet neurons and TMC-expressing neurons on the proboscis project their axons to overlapping areas in the subesophageal zone (SEZ) in the brain (*Zhang et al., 2016*) and we confirmed this by using an independently generated tool to label sweet neurons (*Figure 5A and B*;

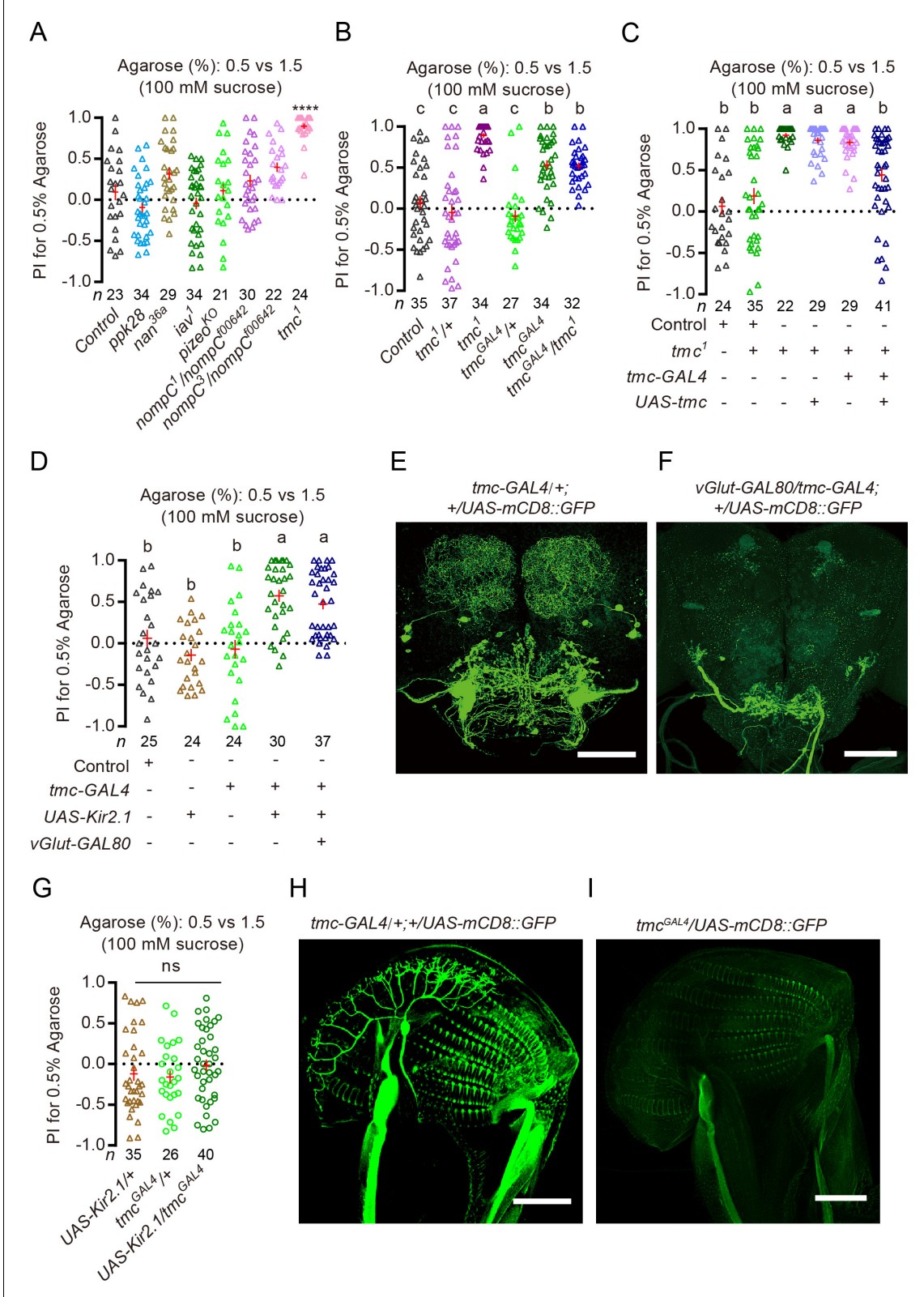

**Figure 4.** Sucrose-induced indifference to substrate of different hardness requires TMC and TMC-expressing neurons. (**A**) PI (for 0.5% agarose) of different channel mutants in a sucrose + 0.5% vs. sucrose + 1.5% two-choice assay. ****p<0.0001; Mann–Whitney test, compared against control. (**B and C**) PI (for 0.5 agarose) of *tmc* mutants and mutants with *tmc* selectively rescued in *tmc-GAL4*-expressing neurons in a sucrose + 0.5% vs. sucrose + 1.5% two-choice assay. *tmc¹* and *tmcᴳᴬᴸ⁴* are two independently generated mutations in *tmc*. Groups that share at least one letter are statistically

*Figure 4 continued on next page*

*Figure 4 continued*

indistinguishable; Kruskal–Wallis test followed by Dunn's multiple comparisons test, p<0.05. Note that *tmc^GAL4* and *tmc-GAL4* are two independently generated *GAL4s*. (D) PI (for 0.5% agarose) of females with *tmc-GAL4*-expressing neurons selectively silenced in the presence and absence of *vGlu-GAL80* in a sucrose + 0.5% vs. sucrose + 1.5% two-choice assay. Groups that share at least one letter are statistically indistinguishable; Kruskal–Wallis test followed by Dunn's multiple comparisons test with p<0.05. (E and F) Processes labeled by *tmc-GAL4* in the brain in the (E) absence and (F) presence of *vGlut-GAL80*. (G) PI (for 0.5% agarose) for females whose TMC-expressing neurons were inhibited by using *tmc^GAL4*, an independently generated *GAL4* for *tmc*, in a sucrose + 0.5% vs. sucrose + 1.5% two-choice assay. ns: not significant; Kruskal–Wallis test followed by Dunn's multiple comparisons test. (H and I) Comparison of expression patterns on the labellum driven by *tmc-GAL4* vs. *tmc^GAL4*.

DOI: https://doi.org/10.7554/eLife.46165.012

The following source data and figure supplements are available for figure 4:

**Source data 1.** Raw neumerical data for *Figure 4*.
DOI: https://doi.org/10.7554/eLife.46165.019

**Figure supplement 1.** TMC-expressing neurons on the labellum are required for sucrose-induced inhibition of discrimination of substrate hardness.
DOI: https://doi.org/10.7554/eLife.46165.013

**Figure supplement 1—source data 1.** Raw numerical data for *Figure 4—figure supplement 1*.
DOI: https://doi.org/10.7554/eLife.46165.014

**Figure supplement 2.** Additional characterizations of *tmc* expression and *tmc* mutant phenotype.
DOI: https://doi.org/10.7554/eLife.46165.015

**Figure supplement 2—source data 1.** Raw numerical data and gel picture for *Figure 4—figure supplement 2*.
DOI: https://doi.org/10.7554/eLife.46165.016

**Figure supplement 3.** Feeding preference of WT and *tmc* mutants.
DOI: https://doi.org/10.7554/eLife.46165.017

**Figure supplement 3—source data 1.** Raw neumerical data for *Figure 4—figure supplement 3*.
DOI: https://doi.org/10.7554/eLife.46165.018

*Figure 5—figure supplement 1A*). Given the close proximity of their projections, we wondered whether axons of sweet neurons may directly contact those of TMC-expressing neurons in the SEZ. To test this idea, we first used the 'conventional' GRASP technique (*Feinberg et al., 2008*) that targeted two halves of the GFP – CD4-spGFP1-10 and CD4-spGFP11 – to the cell membranes of the sweet neurons and the TMC-expressing neurons, respectively. Encouragingly, we detected reconstituted GFP clearly at areas targeted by both sets of axons in the SEZ (*Figure 5—figure supplement 1B and C*). To test whether these putative contacts might be synaptic in nature, we switched to using the syb:GRASP technique (*Macpherson et al., 2015*) where one-half of the GFP was targeted specifically to the presynaptic termini of the sweet neurons and would be exposed to the synaptic cleft only upon vesicle fusion. Again, we were able to detect reconstituted GFP clearly in the SEZ (*Figure 5C*; *Figure 5—figure supplement 1D*), suggesting that sweet neurons can conceivably act as a presynaptic partner to axonal termini of TMC neurons in the SEZ. In contrast, we did not detect any reconstituted GFP signals in the ventral nerve cord (*Figure 5—figure supplement 1E*).

To probe whether the potential axonal–axonal contacts we observed using the GRASP techniques permit the sweet neurons to transmit signal to the TMC-expressing neurons, we next assessed if we could detect a change in Ca$^{2+}$ influx in the axonal termini of TMC neurons when we stimulated the sweet neurons. To do this, we developed an *ex vivo* preparation where we monitored Ca$^{2+}$ influx into axons of TMC-expressing by using the genetically encoded Ca$^{2+}$ indicator GCaMP6s (*Chen et al., 2013*). We first exposed the preparation to sucrose directly, and indeed observed a significant increase in GCaMP signal in the axonal termini of TMC neurons (*Figure 5D and E*). Several additional results suggest that the increase in GCaMP signal we observed was neither caused by movements nor because TMC neurons were intrinsically sugar sensitive. First, exposing the preparations to sorbitol, a sugar that cannot activate *Drosophila* sweet-taste neurons and cannot inhibit discrimination of substrates of different hardness during egg-laying (*Figure 3—figure supplement 1B*), failed to induce a significant increase in GCaMP in the axonal termini of TMC neurons (*Figure 5—figure supplement 2B*). Second, cell bodies of TMC neurons on the proboscis did not respond to sucrose whereas those of sweet neurons did (*Figure 5—figure supplement 2C and D*). Third, addition of sucrose failed to induce as significant an increase in GCaMP in TMC axons from mutant that lacked Gr64f (*Figure 5D and E*); Gr64f is a molecular co-receptor for sugar that has been shown to be important for sugar sensing (*Fujii et al., 2015*; *Yavuz et al., 2014*). Further, we also found that

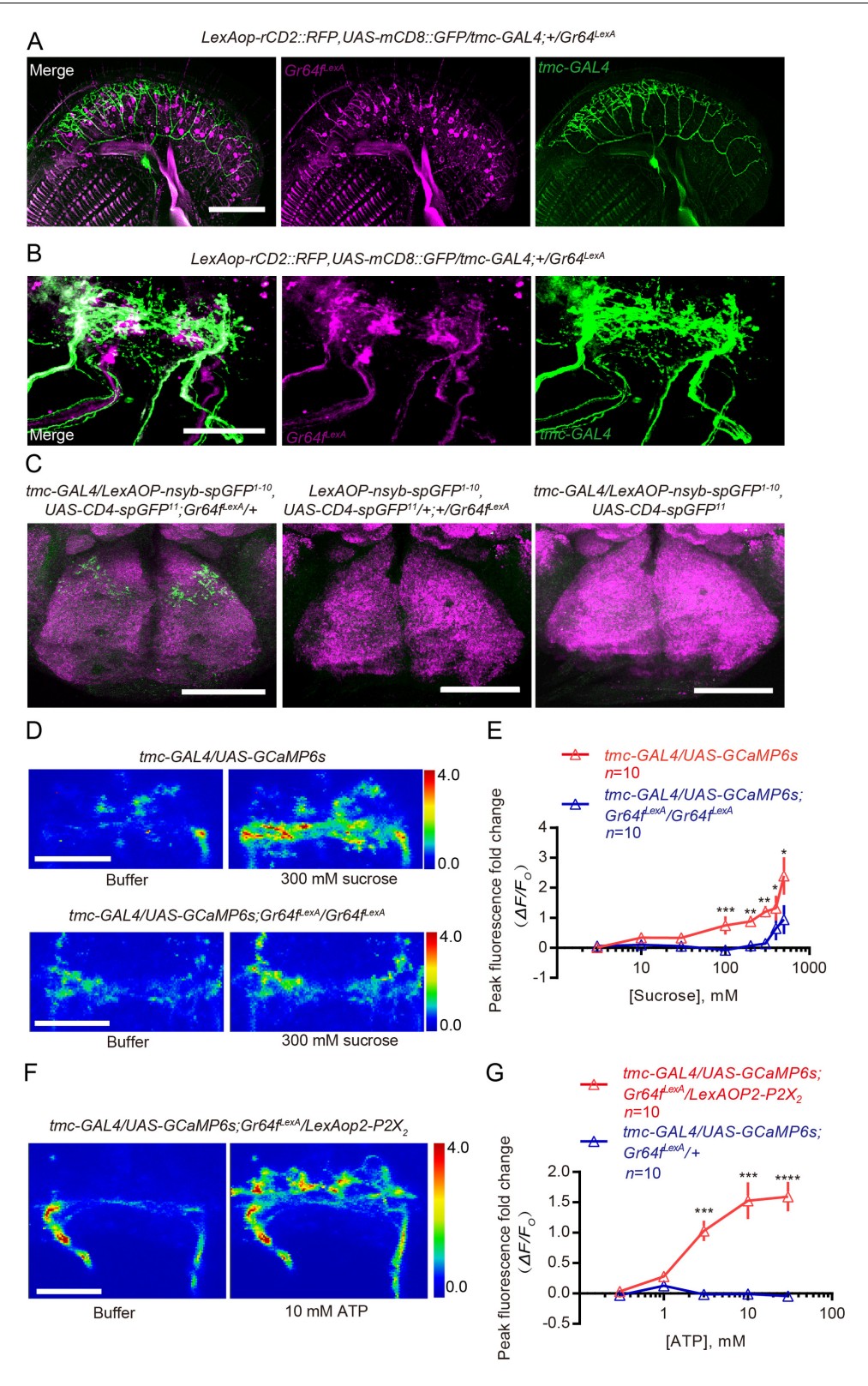

**Figure 5.** Axons of sweet neurons have physical contact with and can signal to axons of TMC-expressing neurons. (**A and B**) Double labeling of the TMC-expressing MD-L neurons and Gr64f-expressing sweet neurons on the labellum (**A**) and in the SEZ (**B**) in the brain. Scale bar: 50 μm. (**C**) syb: GRASP (green) between TMC-expressing MD-L neurons and sweet neurons. These brains were counter-stained with neuropil marker nc82 (magenta). Scale bar: 50 μm. (**D**) Representative images showing buffer- and sucrose-induced changes in CGaMP signal in axon termini of TMC neurons in the SEZ.

*Figure 5 continued on next page*

*Figure 5 continued*

Top: preparation made from WT animals; bottom: preparation made from *Gr64f* mutants (*Gr64f^LexA*). The color scale on the right shows $\Delta F/F$. (**E**) Changes in peak GCaMP intensity ($\Delta F/F_0$) in TMC axons from WT vs. *Gr64f* mutants in response to different concentrations of sucrose. ns: not significant, ***$p<0.001$, **$p<0.01$; Mann–Whitney test. (**F**) Representative images showing buffer- and ATP-induced changes in the GCaMP signal of TMC axons in preparations made from animals that overexpressed P2X$_2$ in Gr64f-expressing sweet neurons. (**G**) Changes in peak GCaMP intensity ($\Delta F/F_0$) in TMC axons from animals with or without P2X$_2$ overexpressed in Gr64f-expressing sweet neurons in response to different concentrations of ATP. ****$p<0.0001$, ***$p<0.001$; Mann–Whitney test.

DOI: https://doi.org/10.7554/eLife.46165.020

The following source data and figure supplements are available for figure 5:

**Source data 1.** Raw numerical data for *Figure 5*.

DOI: https://doi.org/10.7554/eLife.46165.024

**Figure supplement 1.** Axons of TMC-expressing neurons are in contact with those of sweet neurons in the SEZ.

DOI: https://doi.org/10.7554/eLife.46165.021

**Figure supplement 2.** Axons of TMC-expressing neurons respond to sucrose but such a response is not intrinsically derived and is diminished in the absence of TMC.

DOI: https://doi.org/10.7554/eLife.46165.022

**Figure supplement 2—source data 1.** Raw numerical data for *Figure 5—figure supplement 2*.

DOI: https://doi.org/10.7554/eLife.46165.023

sucrose-induced GCaMP increase in axons of TMC neurons was reduced in a *tmc* mutant background (*Figure 5—figure supplement 2A*), suggesting that strong increase in Ca$^{2+}$ influx in termini of TMC axons requires co-activation of sweet neurons and input from TMC channels.

Lastly, to confirm independently that activation of sweet neurons can increase Ca$^{2+}$ influx into axons of TMC neurons, we selectively expressed in sweet neurons the ATP-gated P2X$_2$ channel, a channel that is not expressed endogenously in flies and thus can be used for 'chemogenetic activation' of specific neurons of interest in flies (*Lima and Miesenböck, 2005*; *Yao et al., 2012*). We found that stimulating the P2X$_2$-expressing sweet neurons with ATP also triggered a significant increase in GCaMP signal in axonal termini of the TMC neurons (*Figure 5F and G*). In contrast, exposing control animals that did not express P2X$_2$ in their sweet neurons to the same levels of ATP did not produce such an increase (*Figure 5F and G*; *Figure 5—figure supplement 2E and F*).

Taken together, our collective results showed that functional sweet neurons and TMC neurons were both required for sucrose-induced inhibition of discrimination of substrates of different hardness. Further, axons of sweet neurons physically contacted those of TMC neurons in the brain and stimulation of sweet neurons increased Ca$^{2+}$ influx into axonal termini of TMC neurons. Thus, one mechanism by which sucrose-activated sweet neurons can act to inhibit discrimination of substrates of different hardness during egg-laying is by directly enhancing the output of TMC-expressing mechanosensitive neurons.

## Discussion

In this work, we showed that activation of sweet neurons by sucrose can promote *Drosophila* females to become indifferent between two substrates of different hardness during egg-laying, and that such sucrose-induced indifference required input from the TMC-expressing mechanosensitive neurons on the labellum. Specifically, we showed that *Drosophila* females generally preferred the softer substrate for egg-laying in a two-choice assay when both options were sugar free, but their preference for the softer substrate reduced significantly when both options contained 100 mM sucrose. Such sugar-induced indifference between substrates of different hardness depended on functional molecular sugar receptors and sweet neurons as well as, interestingly, functional TMC channel and TMC-expressing mechanosensitive neurons. Further, our anatomical-labeling and Ca$^{2+}$-imaging results showed that axons of sweet neurons directly contacted those of TMC-expressing neurons in the brain and that depolarizing the sweet neurons increased Ca$^{2+}$ influx into axon termini of TMC neurons. Thus, such axon-axon contacts provide an anatomical basis for sweet neurons to directly modulate the output of TMC neurons in the brain. Together, these findings suggest that, during egg-laying site selection, activation of sweet neurons can act to inhibit discrimination of substrates of different hardness by enhancing the output of TMC neurons directly. Our results thus

demonstrate a novel means by which *Drosophila* integrate specific chemosensory and mechanosensory properties of two competing substrates when evaluating them during a simple decision-making task. However, it is worth pointing out that the mechanism we described may not be the only path by which sweet neurons can act to modify discrimination of substrate hardness during egg-laying site selection. First, input from tarsi and antennae played a role, too. While we did detect *tmc* transcripts on them (*Figure 4—figure supplement 2A*), it is unclear whether *tmc*-expressing neurons on these structures (that were missed by the *tmc-GAL4*) have the same interaction with sweet neurons as the ones on the labellum. Second, while the function of *tmc-GAL4*-expressing neurons was required for sucrose to dampen hardness discrimination, we were unable to ascertain that direct artificial activation of these neurons was *sufficient* to do so in the absence of sucrose as such activation severely reduced females' egg-laying rate (*Figure 4—figure supplement 1G–I*). Thus, one important next task is to identify the relevant mechanosensitive input from tarsi and antennae and assess how information they relay might be modulated by activation of sweet neurons during egg-laying site selection.

A second point that is worth discussing is whether our conclusions are compatible with findings from previous reports. While our results suggest that sweet neurons can act to potentiate the output of TMC neurons via axon-axon interaction, two recent studies have shown that activation of mechanosensitive neurons can inhibit the output of sweet neurons. Specifically, *Zhang et al. (2016)* have shown that activation of TMC neurons can inhibit PER, a motor response triggered by activation of sweet neurons. Further, *Jeong et al. (2016)* have shown that Nanchung-expressing neurons can inhibit PER and that axons of Nanchung-expressing neurons form inhibitory synapses with axons of sweet neurons. We proposed our conclusions are not incompatible with these earlier reports. First, it is conceivable that axons of mechanosensitive neurons and sweet neurons can have two distinct types of interactions: presynaptic inhibition from mechanosensitive neurons to sweet neurons as well as presynaptic facilitation from sweet neurons to TMC neurons. Second, while 100 mM sucrose may facilitate TMC neurons less when flies were sampling 1.5% agarose than on 0.5% agarose (taking into account that sweet neurons should be suppressed more on 1.5% agarose than on 0.5% agarose), this should reduce the difference in perceived hardness of 0.5% and 1.5% agarose substrates, thus not inconsistent with what we have seen. Moreover, it is unclear whether 0.5% and 1.5% agarose exerted very different levels of suppression on output of sweet neurons in our task. For example, *Jeong et al. (2016)* showed that 0.2% vs. 2% agarose had significantly different impacts on feeding preference for 0.5 mM vs. 1 mM sucrose, however, the concentration of sucrose we used was 100 mM. For these reasons, we currently favor the idea that our conclusions expand the view of the relationship between sweet neurons and mechanosensitive neurons provided by the previous studies.

Another point worth discussing after comparing our work with previous reports is that flies appeared to use two different sensory mechanisms to discriminate substrates of different hardness during feeding and egg-laying, even though they generally preferred the softer substrate in both tasks. Previous studies have shown that flies rely on TMC, Nan, and NompC channels and two specific groups of labellum sensory neurons that express these channels to discriminate substrates of different hardness during feeding (*Jeong et al., 2016*; *Zhang et al., 2016*; *Sánchez-Alcañiz et al., 2017b*). In contrast, our results showed that neither these channels nor these neurons were essential for flies to discriminate substrates of different hardness during egg-laying. More curiously, our results suggest input from mechanosensitive neurons on the labellum (as well as possibly ones on antennae and tarsi) can act to inhibit discrimination of substrates of different hardness during egg-laying. This conclusion is supported in part by the observations that animals without intact labellum or functional TMC-expressing neurons on the labellum showed enhanced discrimination in the presence of sucrose during egg-laying. In contrast, *tmc* mutants did not discriminate substrates of different hardness well for feeding when given the exact same choices (*Figure 4—figure supplement 3F*). The striking difference in the requirement of labellum and TMC on substrate hardness discrimination during feeding and egg-laying raises the question of what are the identities of the specific sensory neurons that promote discrimination of substrate hardness during egg-laying. The totality of our current results are consistent with a very *tentative model* (see *Figure 6*) that *Drosophila* likely use some as-yet-unidentified mechanosensitive neurons on their ovipositor to sense and discriminate substrates of different hardness. We based this tentative model on the following reasons. First, ovipositor is known to possess mechanosensitive neurons (*Sánchez-Alcañiz and Benton, 2017a*;

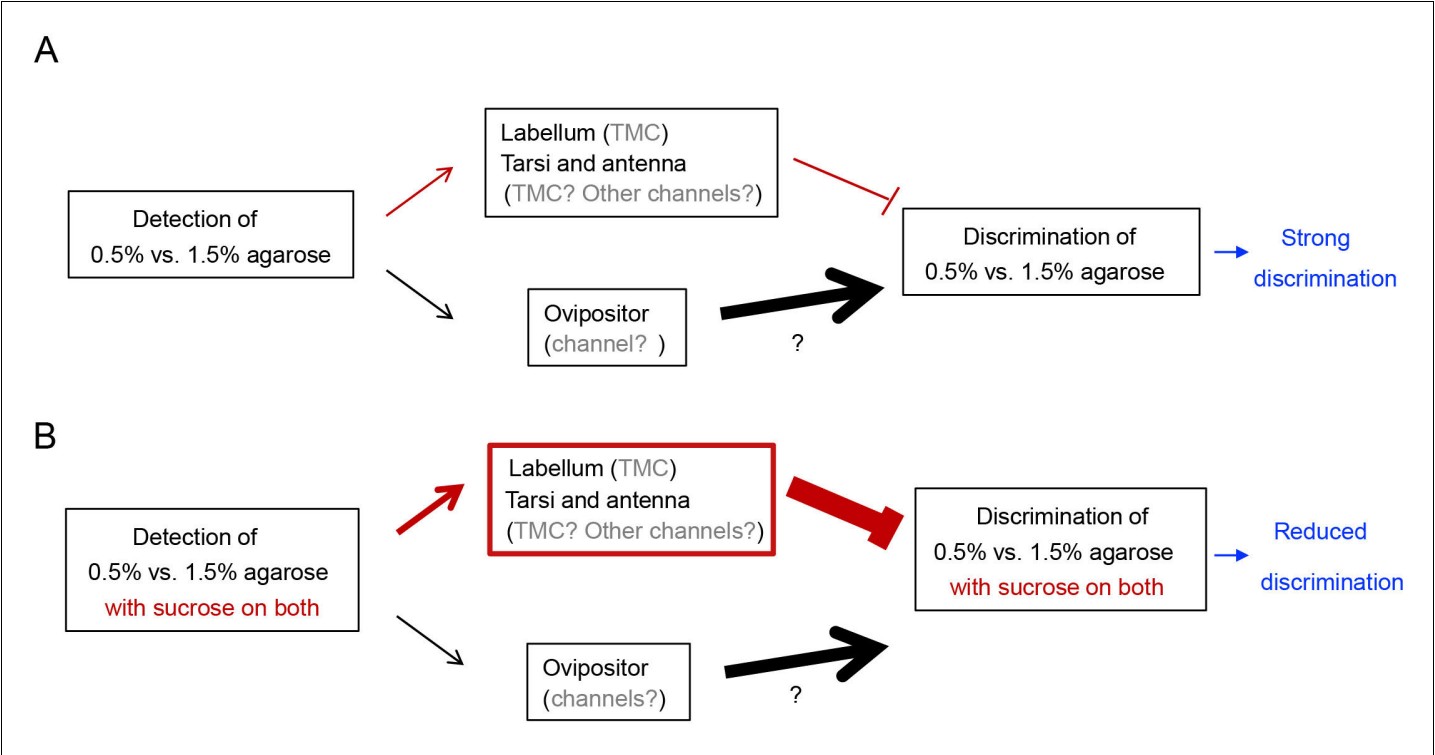

**Figure 6.** A tentative model explaining how discrimination of egg-laying substrates of different hardness is regulated by mechanosensory neurons on different appendages. (A) In the absence of sucrose, detection of hardness of egg-laying substrates by mechanosensory neurons on the ovipositor promotes discrimination whereas detection of hardness of substrates by mechanosensory neurons on the tarsi and labellum inhibits discrimination. Moreover, animals discriminate the plain 0.5% vs. 1.5% agarose well because the contribution from the mechanosensory neurons on the ovipositor dominates over that from the tarsi and labellum in this decision. (We note that while *tmc* transcripts are present on tarsi, labellum, and antenna, inhibition of hardness discrimination in the absence of sucrose may be promoted by mechanosensitive channels that we have not identified in this work. Further, we have only indirect evidence that supports the idea that the specific mechanosensitive neurons critical for egg-laying substrate discrimination are present on the ovipositor.) (B) In the presence of sucrose, however, contribution from the mechanosensory neurons on the labellum, tarsi, and antenna increases, partly because output of TMC neurons on the labellum can be enhanced by sucrose-induced activation of sweet neuron. Consequently, animals discriminate the two sweet substrates less well due to enhanced input from mechanosensitive neurons that inhibit discrimination. (We note that TMC-expressing neurons on the tarsi and antenna may contribute to sucrose-induced inhibition, too, given that *tmc* transcripts were present on both. However, their precise relationship with sweet neurons is not known. Further, some mechanosensitive channels that we have not identified in this work may contribute, too).

DOI: https://doi.org/10.7554/eLife.46165.025

*Stocker, 1994*; *Newland and Burrows, 1994*); second, flies have been shown to actively probe the substrates with their ovipositor prior to depositing each egg (*Yang et al., 2008*); third and most importantly, animals that lacked the a significant portion of virtually all other appendages (e.g. labellum, tarsi, wings) but had intact ovipositor were still capable of discriminating substrates of different hardness. Thus, another important next task is to identify the mechanosensitive neurons on the ovipositor – or possibly on other body parts – that are critical for discriminating substrate hardness during egg-laying and the central targets of these neurons. Identities of these neurons will provide a much-needed molecular and anatomical basis to start elucidating how texture discrimination and substrate selection during egg-laying site selection is enabled and modulated.

Lastly, what is the potential advantage in allowing sugar detection to inhibit discrimination of egg-laying substrates of different hardness? Strong selectiveness likely costs effort and delays emergence of progenies. Thus, when deciding between two competing substrates that do not differ significantly in values, it might be more advantageous for flies to deposit their eggs on both. In our experiments, difference in values between the plain 0.5% agarose and the plain 1.5% agarose maybe relatively small because while flies preferred the 0.5% agarose over the 1.5% agarose in the two-choice assay, they laid comparable numbers of eggs on them when each was presented in single-

choice assays. Thus, the presence of high concentration of sucrose in both substrates may further reduce their differences in values, thereby largely eliminating flies' soft preference. (However, it is worth noting that we favored the idea that adding sucrose to the 0.5% and 1.5% agarose substrates may equalize their values by dampening them, at least in the context of our regular assays. This is because in our regular assays, adding sucrose to an agarose substrate *reduces* as opposed to increases its value: while flies readily accepted the sucrose-containing substrate for egg-laying, they consistently preferred the plain one when given a choice between a plain one and a sucrose-containing one to choose from [*Yang et al., 2008*; *Yang et al., 2015*]) (*Figure 4—figure supplement 1F*). Finally, from an evolutionary point of view, we proposed that allowing sweet neurons to directly enhance the output of mechanosensitive neurons that can inhibit hardness discrimination during egg-laying may provide a neural substrate for different species to adopt different texture selectivity. For example, in contrast to *Drosophila melanogaster*, the fruit pest *Drosophila suzukii* is more receptive to lay eggs on harder substrates and attack both ripe (harder) and rotten (softer) fruits (*Karageorgi et al., 2017*; *Atallah et al., 2014*). It may be interesting to test whether modifications of the structure and function of sweet and TMC neurons, and/or the connection between them, contribute to *Drosophila suzukii's* acceptance of harder substrate during egg-laying.

# Materials and methods

**Key resources table**

| Reagent type (species) or resource | Designation | Source or reference | Identifiers | Additional information |
|---|---|---|---|---|
| Genetic reagent (D. melanogaster) | $w^{1118}$ | Bloomington Drosophila Stock Center | BDSC: 3605; RRID:BDSC_3605 | |
| Genetic reagent (D. melanogaster) | ppk28 | PMID:21515576 | BDSC_33559; RRID:BDSC_33559 | |
| Genetic reagent (D. melanogaster) | $nan^{36a}$ | *Kim et al., 2003*; *Gong et al., 2004* | BDSC_24902; RRID:BDSC_24902 | |
| Genetic reagent (D. melanogaster) | $iav^1$ | *Kim et al., 2003*; *Gong et al., 2004* | DGGR:101174; RRID: DGGR_101174 | |
| Genetic reagent (D. melanogaster) | $nompC^{f00642}$ | *Sun et al., 2009* | FlyBase Cat# FBst1016369; RRID: FlyBase_FBst1016369 | |
| Genetic reagent (D. melanogaster) | $nompC^1$ | *Walker et al., 2000* | BDSC_42260; RRID:BDSC_42260 | |
| Genetic reagent (D. melanogaster) | $nompC^3$ | *Walker et al., 2000* | BDSC_42258; RRID:BDSC_42258 | |
| Genetic reagent (D. melanogaster) | $Piezo^{KO}$ | *Kim et al., 2012* | BDSC_58770; RRID:BDSC_58770 | |
| Genetic reagent (D. melanogaster) | $tmc^{pb}$ | Bloomington Drosophila Stock Center | BDSC_18483; RRID:BDSC_18483 | |
| Genetic reagent (D. melanogaster) | $tmc^{MI02041}$ | Bloomington Drosophila Stock Center | BDSC_35958; RRID:BDSC_35958 | |
| Genetic reagent (D. melanogaster) | $tmc^1$ | *Zhang et al., 2016* | BDSC_66556; RRID:BDSC_66556 | |
| Genetic reagent (D. melanogaster) | UAS-tmc/Cyo; tmc1/TM6B | *Zhang et al., 2016* | BDSC_66560; RRID:BDSC_66560 | |
| Genetic reagent (D. melanogaster) | tmc-GAL4/Cyo; MKRS/TM6B | *Zhang et al., 2016* | BDSC_66557; RRID:BDSC_66557 | |
| Genetic reagent (D. melanogaster) | $nompC-LexA^{ll}$ | *Shearin et al., 2013* | BDSC_52240; RRID:BDSC_52240 | |

*Continued on next page*

*Continued*

| Reagent type (species) or resource | Designation | Source or reference | Identifiers | Additional information |
|---|---|---|---|---|
| Genetic reagent (*D. melanogaster*) | *nompC-LexA^III* | *Shearin et al., 2013* | BDSC_52241; RRID:BDSC_52241 | |
| Genetic reagent (*D. melanogaster*) | *R41E11-GAL4* | Bloomington Drosophila Stock Center | BDSC_50131; RRID:BDSC_50131 | |
| Genetic reagent (*D. melanogaster*) | *Gr5a-GAL4* | *Weiss et al., 2011* | BDSC_57592; RRID:BDSC_57592 | |
| Genetic reagent (*D. melanogaster*) | *Gr64f-GAL4* | *Weiss et al., 2011* | BDSC_57668; RRID:BDSC_57668 | |
| Genetic reagent (*D. melanogaster*) | *Gr66a-GAL4* | *Dunipace et al., 2001* | BDSC_57670; RRID:BDSC_57670 | |
| Genetic reagent (*D. melanogaster*) | *TubP-GAL80^ts/ Cyo;TM2/TM6B* | Bloomington Drosophila Stock Center | BDSC_7019; RRID:BDSC_7019 | |
| Genetic reagent (*D. melanogaster*) | *vGlut-GAL80/ Cyo;TM2/TM6B* | *Bussell et al., 2014* | BDSC_58448; RRID:BDSC_58448 | |
| Genetic reagent (*D. melanogaster*) | *UAS-GCaMP6s* | *Chen et al., 2013* | BDSC_77131; RRID:BDSC_77131 | |
| Genetic reagent (*D. melanogaster*) | *UAS-TNT* | *Sweeney et al., 1995* | BDSC_28838; RRID:BDSC_28838 | |
| Genetic reagent (*D. melanogaster*) | *UAS-TNT^IMP* | *Sweeney et al., 1995* | BDSC_28840; RRID:BDSC_28840 | |
| Genetic reagent (*D. melanogaster*) | *LexAop-rCD2::RFP, UAS-mCD8::GFP/ Cyo;TM3/TM6B* | Bloomington Drosophila Stock Center | BDSC_67093; RRID:BDSC_67093 | |
| Genetic reagent (*D. melanogaster*) | *LexAOP2-P2 × 2* | *Gou et al., 2014* | BDSC_76030; RRID:BDSC_76030 | |
| Genetic reagent (*D. melanogaster*) | *LexAOP-mCD4 -spGFP11, UAS-mCD4-spGFP1-10* | *Gordon and Scott, 2009* | | |
| Genetic reagent (*D. melanogaster*) | *LexAOP-nSyb -spGFP1-10, UAS-CD4-spGFP11; MKRS/TM6B* | *Macpherson et al., 2015* | BDSC_64315; RRID:BDSC_64315 | |
| Genetic reagent (*D. melanogaster*) | *UAS-CsChrimson* | *Klapoetke et al., 2014* | BDSC_55136; RRID:BDSC_55136 | |
| Genetic reagent (*D. melanogaster*) | *UAS-dTrpA1* | *Hamada et al., 2008* | BDSC_26263; RRID:BDSC_26263 | |
| Genetic reagent (*D. melanogaster*) | *Gr5a^LexA* | *Yavuz et al., 2014; Fujii et al., 2015* | Flybase: FBal0304286 | |
| Genetic reagent (*D. melanogaster*) | *Sugar-blind (Δ8Gr^sugar/Δ8Gr^suga)* | *Yavuz et al., 2014; Fujii et al., 2015* | Flybase: FBrf0228945 | |
| Genetic reagent (*D. melanogaster*) | *Gr64a^GAL4* | *Yavuz et al., 2014; Fujii et al., 2015* | Flybase: FBal0304287 | |
| Genetic reagent (*D. melanogaster*) | *Gr64f^LexA* | *Yavuz et al., 2014; Fujii et al., 2015* | Flybase: FBal0304291 | |
| Genetic reagent (*D. melanogaster*) | *UAS-Kir2.1* | Bloomington Drosophila Stock Center | Flybase: FBtp0125506 | |
| Genetic reagent (*D.melanogaster*) | *tmc^GAL4* | *Guo et al., 2016* | Flybase: FBal0321088 | |
| Chemical, compound, drug | sucrose | Sigma-Aldrich | #S0389 | |
| Chemical, compound, drug | fructose | Sigma-Aldrich | #F0127 | |
| Chemical, compound, drug | glucose | Sigma-Aldrich | #G8270 | |

*Continued on next page*

*Continued*

| Reagent type (species) or resource | Designation | Source or reference | Identifiers | Additional information |
|---|---|---|---|---|
| Chemical, compound, drug | trehalose | Sigma-Aldrich | #T5251 | |
| Chemical, compound, drug | arabinose | Sigma-Aldrich | #A3131 | |
| Chemical, compound, drug | sorbitol | Sigma-Aldrich | #W302902 | |
| Chemical, compound, drug | caffeine | Sigma-Aldrich | #C0750 | |
| Chemical, compound, drug | all trans-Retinal | Sigma-Aldrich | #R2500 | |
| Chemical, compound, drug | acetic acid | EMD Millipore | #AX0073 | |
| Chemical, compound, drug | agarose | Invitrogen | #16500–100 | |
| Chemical, compound, drug | adenosine 5'-triphosphate (ATP) | Amersham Biosciences | #27-1006-01 | |
| Antibody | anti-brp (mouse monoclonal) | Developmental Studies Hybridoma Bank | #nc82; RRID:AB_2314866 | (1:50) |
| Antibody | anti-GFP (mouse monoclonal) | Thermo Fisher Scientific, Waltham, MA | #A-11120, RRID:AB_221568 | (1:200) |
| Antibody | anti-GFP (rabbit polyclonal) | Thermo Fisher Scientific, Waltham, MA | #A-11122, RRID:AB_221569 | (1:1000) |
| Antibody | anti-RFP (rabbit polyclonal) | Rockland | #600-401-379, RRID:AB_2209751 | (1:500) |
| Antibody | Donkey Alexa 488 anti-rabbit secondaries | Thermo Fisher Scientific, Waltham, MA | #R37118, RRID:AB_2556546 | (1:500)) |
| Antibody | Donkey Alexa 488 anti-mouse secondaries | Thermo Fisher Scientific, Waltham, MA | #R37114, RRID:AB_2556542 | (1:500) |
| Antibody | Donkey Alexa 594 anti-rabbit secondaries | Thermo Fisher Scientific, Waltham, MA | #R37119, RRID:AB_2556547 | (1:500) |
| Antibody | Donkey Alexa 594 anti-mouse secondaries | Thermo Fisher Scientific, Waltham, MA | #R37115, RRID:AB_2556543 | (1:500) |
| Software, algorithm | ImageJ | PMID: 22930834 | https://imagej.nih.gov/ij; RRID:SCR_003070 | |
| Software, algorithm | Prism 6 | Graphpad | RRID:SCR_002798 | |
| Software, algorithm | Photoshop | Adobe | RRID:SCR_014199 | |
| Software, algorithm | Illustrator | Adobe | RRID:SCR_010279 | |
| Software, algorithm | Ctrax | *Branson et al., 2009* | | |
| Software, algorithm | Custom MATLAB code | *Yang et al., 2015* | https://github.com/ulrichstern/yanglab-ctrax | |

## Fly stocks

All fly stocks were raised in a standard cornmeal–molasses–agar medium maintained at 25 °C, 60% humidity, and a 12 hr:12 hr light:dark circadian control. We used $w^{1118}$ flies as the wild-type strain. The following lines were used in this study: *ppk28* (*Cameron et al., 2010*); *nan$^{36a}$* (*Kim et al., 2003*; *Gong et al., 2004*); *iav$^1$* (*Kim et al., 2003*; *Gong et al., 2004*); *nompC$^{f00642}$* (*Sun et al., 2009*); *nompC$^1$* (BL-42260) (*Walker et al., 2000*); *nompC$^3$* (BL-42258) (*Walker et al., 2000*); *Piezo$^{KO}$* (BL-58770) (*Kim et al., 2012*); *tmc$^{pb}$* (BL-18483); *tmc$^{MI02041}$* (BL-35958); *tmc$^1$* (BL-66556) (*Zhang et al., 2016*); *UAS-tmc/Cyo; tmc$^1$/TM6B* (BL-66560) (*Zhang et al., 2016*); *tmc-GAL4/Cyo; MKRS/TM6B* (BL-66557); *nompC-LexA$^{II}$* (BL52240), *nompC-LexA$^{III}$* (BL52241) (*Shearin et al., 2013*); *R41E11-GAL4* (BL-50131); *Gr5a-GAL4* (BL-57592) (*Weiss et al., 2011*); *Gr64f-GAL4* (BL-57668) (*Weiss et al., 2011*); *Gr66a-GAL4* (BL-57670) (*Dunipace et al., 2001*); *TubP-GAL80$^{ts}$/Cyo;TM2/TM6B* (BL-7019); *vGlut-*

GAL80/Cyo;TM2/TM6B (BL-58448) (*Bussell et al., 2014*); UAS-GCaMP6s (BL-77131) (*Chen et al., 2013*); UAS-TNT (BL-28838) (*Sweeney et al., 1995*); UAS-TNT$^{IMP}$ (BL-28840) (*Sweeney et al., 1995*); LexAop-rCD2::RFP, UAS-mCD8::GFP/Cyo;TM3/TM6B (BL-67093); LexAOP2-P2 $\times$ $_2$ (*Gou et al., 2014*); LexAOP-mCD4-spGFP$^{11}$, UAS-mCD4-spGFP$^{1-10}$ (*Gordon and Scott, 2009*), LexAOP-nSyb-spGFP$^{1-10}$, UAS-CD4-spGFP$^{11}$; MKRS/TM6B (BL-64315) (*Macpherson et al., 2015*). UAS-CsChrimson (*Klapoetke et al., 2014*), UAS-dTRPA1 (*Hamada et al., 2008*), Gr5a$^{LexA}$, sugar blind, Gr64a$^{GAL4}$, and Gr64f$^{LexA}$ were gifts from Dr. Hubert Amrein (*Fujii et al., 2015*; *Yavuz et al., 2014*). UAS-Kir2.1 and tmc$^{GAL4}$ (*Guo et al., 2016*) were from Drs. Gwyneth Card and Yuh-Nung Jan, respectively.

## Chemicals

The following chemicals were purchased from Sigma Aldrich (Saint Louis, MO, USA): sucrose (Cat# S0389), fructose (Cat# F0127), glucose (Cat# G8270), trehalose (Cat# T5251), arabinose (Cat# A3131), sorbitol (Cat# W302902), caffeine (Cat# C0750), and *all-trans*-retinal (Cat#R2500). Acetic acid (Cat# AX0073) was purchased from EMD Millipore (EMD Millipore, Billerica, MA). Agarose and adenosine 5′-triphosphate (ATP) were purchased from Invitrogen (Cat# 16500–100, Carlsbad, CA) and Amersham Biosciences (Cat# 27-1006-01, Piscataway, NJ), respectively.

## Behavior assays

Unless otherwise mentioned, all behavior assays were performed with females in darkness with temperature and humidity controlled at 25°C and 60%, respectively.

### Preparation of females to be assayed

The experimental flies were prepared as previously described (*Yang et al., 2015*; *Gou et al., 2014*). Briefly, 20 to 30 females of the appropriate genotypes and 10 to 15 males of mixed genotypes were gathered into a single food vial that was supplied with wet yeast paste. After 4-8 days, the food in the vial became very chewed up by the larvae, and at this point females were well fed but deprived of egg-laying. Thus, they were ready to lay eggs when placed in our egg-laying apparatus (*Yang et al., 2015*). We usually let females lay eggs overnight (~14 hr). Groups to be compared were always run in parallel. Note that regardless of the assays we performed (single-choice vs. two-choice vs. large arenas), we always assayed at the level of single animals — each data point on a graph denotes the outcome (number of eggs laid or PI) of a single female.

### Two-choice egg-laying assay in regular arenas

For two-choice assays, individual flies were placed in custom-made transparent Plexiglas chambers. Detailed information of the chambers was described before (*Gou et al., 2016*). For conditional inactivation experiments using UAS-Kir2.1 and tub-GAL80$^{ts}$, all flies were maintained at 22°C except for the shift in the experimental group (where the temperature increased to 31°C for 2–3 days before the assay to inactivate GAL80$^{ts}$). For optogentic activation experiments, flies were maintained similarly as described before except that immediately after eclosion, flies to be assayed were put into foil-covered food vials that contained 0.2 mM *all-trans*-retinal until ready for experiments. When ready, flies were then loaded into egg-laying apparatus illuminated with or without red LED light from above and allowed to lay eggs for ~14 hr (*Guntur et al., 2017*). For thermogenetic activation experiments, flies were raised and egg-laying deprived (as described before) at 22°C, and then placed at either control (22°C) or experimental temperatures (30°C) for ~14 hr for egg-laying. For *two-choice assay in large arenas*, we followed the same protocol for preparing the flies but used a custom-apparatus with arenas that are ~8X the size of that of regular arenas.

### Egg-laying rate assay (single-choice assay)

For this assay, individual flies were put into two-choice chambers but we loaded the same concentration of agarose onto the two sides of each arena. For all these assays, we took pictures of the results when the assays ended, and manually counted the eggs.

## Positional preference assay

To track the positions of females, we mounted four webcams (Microsoft LifeCam Cinema) on top of the egg-laying apparatus using a custom-built holder. Females and chambers were prepared as described before (*Zhu et al., 2014*; *Stern et al., 2015a*; *Stern et al., 2015b*). However, we recorded behaviors of egg-laying females for 30 min only. We used CamUniversal software for video acquisition, Avidemux software for video conversion, and the open-source tracking software Ctrax (*Branson et al., 2009*) for tracking. Individual egg-laying events in the videos were manually annotated. To analyze the Ctrax-generated trajectories, we used custom MATLAB and Python code (*Stern et al., 2015a*).

## Feeding preference assay

We used our regular-sized egg-laying apparatus to conduct food preference assay. Briefly, for each arena, 100 mM sucrose-containing 0.5% of agarose was placed on one trough and 100 mM sucrose-containing 1.5% agarose was placed on the other. The agarose was mixed with either blue dye (Erioglaucine disodium salt, Solarbio, Cat# E8500) or a red dye (Sulforhodamine B, Solarbio, catalog Cat# S6080). 2–4 day-old females WT and mutant flies were wet-starved for 24 hr and then loaded individually into each arena and allowed to feed in darkness for 90 min. We then examined the colors of the abdomens using a dissecting microscope after feeding was terminated. The food preference index (PI) for the 0.5% agarose (red color) was as follows: $PI = [(N_{red} + 0.5\ N_{purple}) - (N_{blue} + 0.5\ N_{purple})]/(N_{red} + N_{blue} + N_{purple})$. $N_{red}$, $N_{blue}$, and $N_{purple}$ indicated the number of flies with red, blue and purple abdomens, respectively.

## Egg-laying substrate hardness measurements

First, 40 ml of agarose gel (0.5% or 1.5% agarose with or without 100 mM sucrose) was prepared in a 60 mm$^2$ diameter dish (Nunc, Denmark). Agarose hardness, expressed as the maximum force (N), was measured at room temperature with a TA.XT plus texture analyzer (Stable Micro Systems, Goldaming, Surrey, UK) using a 50 mm diameter flat plunger (SMS P/50). The instrument settings were as follows: pre-test speed: 4.0 mm/s; test speed: 2.0 mm/s; post-test speed: 5.0 mm/s; strain: 40%; and trigger force: 5 g. The TPA analysis was carried out at ambient temperature (25˚C). Texture Expert Exceed version 2.5 (Stable Micro Systems) was used for data collection and calculations.

## RT-PCR

Different tissue samples, including the tarsi, antenna, foreleg, proboscis (as positive control) were dissected from 300 female adults (at 3 days after eclosion). Total RNA was isolated from tissue samples using TRIzol Reagent (Invitrogen, Carlsbad, CA) by following the manufacturer's instructions. First-strand cDNA was synthesized with First-Strand cDNA Synthesis kit (Tiangen, Beijing, China) using an oligo(dT)18 primer and a 1 µg total RNA template in a 20 µl reaction, following the manufacturer's protocol. We used I-5 2 x High-Fidelity Master Mix to conduct RT-PCR (Thermo Fisher Scientific Inc, Waltham, MA). The primers used were as follows. *rp49* (control for normalization): forward primer, 5'- GACCATCCGCCCAGCATACAG −3'; reverse primer: AATCTCCTTGCGCTTC TTGGAGGAG. *tmc*: forward primer, 5'- CCTTCTTCCTGCCCATGATA −3'; reverse primer: TAGCGG TTTCTCCTTGCTGT. The RT-PCR products were sequenced and confirmed with BLAST analysis.

## Tissue dissection, staining, and imaging

We generally dissected labellum and brain of four- to eight-day-old females in PBS buffer, and then fixed them in 4% paraformaldehyde in PBS for ~25 min at room temperature. Fixed tissues were processed using standard antibody staining protocol before being mounted with SlowFade diamond (Life Technologies, Carlsbad, CA, USA). Samples were imaged at 20 × or 40 × magnification on Zeiss 700 confocal microscopes (Jena, Germany), and processed with ImageJ. Titers for primary antibodies were as follows: mouse anti-nc82 (Developmental Studies Hybridoma Bank nc82, 1:50), mouse anti-GFP (1:200, A11120, Thermo Fisher Scientific), rabbit anti-GFP (1:1000, A11122, Thermo Fisher Scientific), and rabbit anti-RFP (1:500, 600-401-379, Rockland). Secondary antibodies were Donkey Alexa 488 anti-rabbit, Donkey Alexa 488 anti-mouse, Donkey Alexa 594 anti-rabbit, and Donkey Alexa 594 anti-mouse (1:500, R37118, R37114, R37119, R37115, Thermo Fisher Scientific).

For GRASP, we did not stain the samples with anti-GFP, and imaged the fixed and washed samples directly.

## Calcium imaging

Brains or labellum of 4- to 10-day-old female flies expressing GCaMP6s were dissected and mounted posterior side down onto the center of a customized imaging chamber (*Gou et al., 2014*; *Guntur et al., 2015*). The cuticle and connective tissue covering the SEZ was removed using fine forceps. The proboscis was extended and immobilized with high vacuum grease. The brain and chamber were filled with artificial hemolymph (AHL: 120 mM NaCl, 3 mM KCl, 5 mM TES, 2 mM $CaCl_2$, 1.5 mM trehalose, 10 mM glucose, 10 mM $NaHCO_3$, 4 mM $MgCl_2$, 10 mM HEPES, 10 mM sorbitol), with the final pH adjusted to 7.25 (*Guntur et al., 2017*). GCaMP6 fluorescence was acquired using a Zeiss LSM700 confocal microscope equipped with a 40 × 0.8 NA water immersion objective and the live-series ZEN image acquisition software. Briefly, baseline GCaMP fluorescence was acquired by scanning the cells with a 488 nm laser at 128 × 128 pixels at 8-bit dynamic range. The GCaMP6 signal in ROI acquired before stimulation (baseline) was used as $F_0$, and the peak change in fluorescence after the addition of the test solution was used as F′. For quantification, ΔF/F was calculated as $(F′-F_0)/F_0$ to reflect the changes in GCaMP signals before and after stimulation. For each experiment, we examined GCaMP responses from six to ten animals. The images were acquired and analyzed using the imaging software integrated with the Zeiss LSM 700.

## Statistics

We used the GraphPad Prism6 software package and 'R project' (R Foundation for Statistical Computing, Vienna, Austria, 2005; R-project-org) to graph and statistically analyze data. We tested whether the values were normally distributed using D'Agostino–Pearson omnibus and Shapiro–Wilk normality tests before performing statistical analysis. When data were normally distributed, we used parametric tests; when data were not normally distributed, we used non-parametric tests. All tests were two-tailed. All data are presented as mean ± s.e.m. We also consistently labeled the sample numbers directly on graphs.

### Sample size

Our results were based on analysis of behavior at the level of single animals. Sample sizes were determined prior to experimentation based on both variance and effect sizes. For each experiment, we typically tested 16 single animals per trial and repeated the trial two to three times. All the data are biological replicates.

### Data exclusion

We only excluded data when the number of eggs laid by a female was less than 10 when we determined its preference index (PI). First, regular females generally laid more than 10 eggs in our experimental setting. Second, a low number of eggs would cause the PI to be unreliable: for example, if a female laid only one egg – one on choice A and zero on choice B – its PI for choice A would be 1, but there would be little confidence that this animal truly had this strong a bias.

## Acknowledgements

We thank Drs. Hubert Amrein, Craig Montell, Yuh-Nung Jan, and Gwenyth Card, and the Bloomington *Drosophila* Stock Center (NIH P40OD018537) for reagents. We also thank members of the Neurogenetics Group and the Fly Club at Duke for suggestions, and Yufeng Pan, Jia Huang, Yang Xiang, Charlene Chen, and Ulrich Stern for discussions and comments on the manuscript. SFW was supported by the China Scholarship Council. This work was supported in part by grants to SFW from the National Natural Science Foundation of China (Grant No. 31772205 and 31830075), and in part by grants to CHY from the National Institutes of Health (R01GM100027).

## Additional information

### Funding

| Funder | Author |
|---|---|
| National Institute of General Medical Sciences | Chung-Hui Yang |
| National Natural Science Foundation of China | Shun-Fan Wu<br>Ya-Long Ja<br>Yijie Zhang |

The funders had no role in study design, data collection and interpretation, or the decision to submit the work for publication.

### Author contributions

Shun-Fan Wu, Conceptualization, Data curation, Formal analysis, Funding acquisition, Investigation, Writing—original draft, Writing—review and editing; Ya-Long Ja, Yi-jie Zhang, Investigation, Methodology; Chung-Hui Yang, Conceptualization, Data curation, Supervision, Funding acquisition, Project administration, Writing—review and editing

### Author ORCIDs

Shun-Fan Wu (iD) https://orcid.org/0000-0003-0096-147X
Chung-Hui Yang (iD) https://orcid.org/0000-0003-2117-3595

### Decision letter and Author response

Decision letter https://doi.org/10.7554/eLife.46165.028
Author response https://doi.org/10.7554/eLife.46165.029

## Additional files

### Supplementary files

• Transparent reporting form
DOI: https://doi.org/10.7554/eLife.46165.026

### Data availability

All data generated or analysed during this study are included in the manuscript and supporting files. Source data files for egg-laying and imaging have been provided for all figures.

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
