## [Decision Letter]

Thank you for submitting your article "Sweet neurons inhibit texture discrimination by signaling TMC-expressing mechanosensitive neurons in *Drosophila*" for consideration by *eLife*. Your article has been reviewed by Ronald Calabrese as the Senior Editor, Kristin Scott as the Reviewing Editor, and two reviewers. The reviewers have opted to remain anonymous.

The reviewers have discussed the reviews with one another and the Reviewing Editor has drafted this decision to help you prepare a revised submission.

Summary:

Wu et al., study how *Drosophila melanogaster* integrates mechanosensory and chemosensory information in the context of egg-laying behavior. The authors report that excitation of sugar neurons directly activates TMC-expressing mechanosensory neurons by means of axo-axonic excitatory transmission, thereby suppressing texture discrimination in an egg-laying assay. Their genetic, behavioral and calcium imaging results point to a model in which strong activation of TMC-expressing MD-L neurons in the labellum inhibits oviposition. In addition, their preliminary observation suggests some unknown mechanosensory inputs from ovipositor in turn promote oviposition. In addition to feeding decision, this study on oviposition adds another nice example of the interplay between gustatory and mechanosensory inputs during fly's decision making. The proposed model of antagonistic regulation of oviposition is potentially interesting.

Essential revisions:

1) The authors did not provide direct evidence that strong activation of the TMC-expressing MD-L neurons inhibits oviposition (Figure 6). If their model is correct, one would predict that optogenetic or thermogenetic activation of TMC-expressing MD-L neurons would mimic the effect of sugar in their oviposition choice assay.

2) Surgical removal of labellum, tarsi, or antenna all restored texture discrimination in the presence of sugar. However, TMC-expressing MD-L neurons are present only in the labellum. The authors need to perform additional experiments to address the contribution of tarsi and antenna to oviposition texture discrimination. As it is, their model does not explain why sugar input from antenna (which is extremely speculative; the Amrein lab never demonstrated any sugar response in the olfactory receptor neurons), for example, can inhibit texture discrimination.

One possible experiment is tissue-specific RT-PCR to see whether TMC is also expressed in the antenna or leg. The study will be strengthened if the authors could provide a more comprehensive model to explain their results shown in Figure 3E, Figure 3—figure supplement 1D and Figure 4A-C.

3) At low sucrose concentrations (5mM) flies still prefer the soft side to lay eggs, but this preference disappears at a high sucrose concentration (100mM). Even if females are non-starved (the authors do not specify this in the methods), it could be that this high sucrose promotes feeding independently of food texture, making the flies lay eggs on either of the two agarose pads. In TMC mutants, flies might not be able to sense the sucrose properly and this would affect the feeding preference. The authors should show whether in the presence of sucrose 100 mM everywhere, feeding plays any role. To test whether TMC neurons are necessary for the proper evaluation of food, the authors could use the same behavioral setup but add for example colorants (blue and red) to each of the different agarose concentrations to see if consumption is the same as wild-type flies.

4) Jeong et al., (2016) showed that activation of labellar mechanosensory neurons expressing Nanchung suppresses sugar neurons by means of axo-axonic inhibitory transmission. In this context, one would expect that sugar neuron transmission is dampened by harder substrates (e.g. 1.5% agarose), which could in principle suppress the synaptic output of sugar neurons. In this context, flies are expected to show preference for softer substrate. Similarly, Zhang et al., (2016) showed that the optogenetic activation of TMC neurons leads to a decrease of sucrose induced PER in flies, implying that the mechanosensory neurons are in fact inhibiting the sugar sensory neurons, not the other way around. However here, the authors show that the sugar neurons are in fact activating TMC neurons. The authors need to perform additional experiments to address this discrepancy or at least provide some explanation in the Discussion section.

5) Schwartz et al., (2012) showed that the positive or negative influence of sugar on oviposition depends on the size of the egg laying chamber. The sugar-induced oviposition avoidance observed by the authors is likely an artifact of the small chamber size. It is unclear whether the interesting observation they made in this study is also influenced by the chamber size. The authors should at least conduct some of their key experiments in larger egg-laying chambers to see whether their conclusion still holds.

---

## [Author Response]

Essential revisions:1) The authors did not provide direct evidence that strong activation of the TMC-expressing MD-L neurons inhibits oviposition (Figure 6). If their model is correct, one would predict that optogenetic or thermogenetic activation of TMC-expressing MD-L neurons would mimic the effect of sugar in their oviposition choice assay.

This is a very good suggestion. However, we were unable to test properly whether optogenetic or thermogenetic activation of TMC neurons inhibits discrimination of substrates of different hardness. This is because both manipulations severely reduced egg-laying rate in our hands (see new Figure 4—figure supplement 1G-I). In the case of optogenetics experiment, we found that *tmc>CsChrimson* flies became severely bloated with water and cannot lay eggs when exposed to red light during egg-laying. In the case of thermogenetic experiment, we found that while *tmc>dTRPA1* flies generally did not lay eggs either at 31°C. We suspect this maybe because some of the *tmc-GAL4* labeled neurons were present on one or more of the internal organs and that artificial activation of these neurons caused the animals to become rather unhealthy. However, we note that out of 116 females we tested, there were 7 “escapers” that did lay ≥ 10 eggs – the minimum threshold we consistently applied when we assessed egg-laying preference of an animal. These animals discriminated 0.5% plain agarose vs. 1.5% plain agarose well, interestingly. This result suggests either TMC neurons were not sufficiently activated in these escapers, or that enhancement of output of TMC neurons is only necessary but not sufficient to dampen discrimination of substrate hardness (and that sweet neurons may recruit additional pathways to dampen hardness discrimination). We have included this possibility in the new Discussion section and modified our tentative model accordingly.

2) Surgical removal of labellum, tarsi, or antenna all restored texture discrimination in the presence of sugar. However, TMC-expressing MD-L neurons are present only in the labellum. The authors need to perform additional experiments to address the contribution of tarsi and antenna to oviposition texture discrimination. As it is, their model does not explain why sugar input from antenna (which is extremely speculative; the Amrein lab never demonstrated any sugar response in the olfactory receptor neurons), for example, can inhibit texture discrimination.One possible experiment is tissue-specific RT-PCR to see whether TMC is also expressed in the antenna or leg. The study will be strengthened if the authors could provide a more comprehensive model to explain their results shown in Figure 3E, Figure 3—figure supplement 1D and Figure 4A-C.

It is indeed curious that severing the antennae or the tarsi enhanced discrimination of substrate of different hardness in the presence of sucrose, too. We took the approach suggested by the reviewers and conducted an RT-PCR experiment to assess whether *tmc* may be present on these two appendages, too. We found that, interestingly, while *tmc-GAL4* did not label neurons on either the tarsi or the antennae, *tmc* transcripts were indeed present on both. (These new results are shown in new Figure 4—figure supplement 2A). Thus, TMC and TMC-expressing neurons on the antennae and/or tarsi may *potentially* act together with TMC neurons on the labellum in dampening hardness discrimination in the presence of sucrose. We have therefore modified the model (see new Figure 6) by adding the presence of *tmc* on antennae and tarsi. But we note that we do not know whether the same axon-axon interaction mechanism we described in the manuscript applies to these “TMC neurons” (missed by *tmc-GAL4*) on the tarsi and antennae. For example, as the reviewers have pointed out, it is not known whether the *Gr64f* driver-labeled neurons on the antennae are truly sugar sensing. Further, we also acknowledged in the model that the presence of *tmc* transcripts on antennae and tarsi did not rule out the possible critical involvement of other mechanosensitive neurons/channels on these structures in dampening hardness discrimination during egg-laying site selection.

3) At low sucrose concentrations (5mM) flies still prefer the soft side to lay eggs, but this preference disappears at a high sucrose concentration (100mM). Even if females are non-starved (the authors do not specify this in the methods), it could be that this high sucrose promotes feeding independently of food texture, making the flies lay eggs on either of the two agarose pads. In TMC mutants, flies might not be able to sense the sucrose properly and this would affect the feeding preference. The authors should show whether in the presence of sucrose 100 mM everywhere, feeding plays any role. To test whether TMC neurons are necessary for the proper evaluation of food, the authors could use the same behavioral setup but add for example colorants (blue and red) to each of the different agarose concentrations to see if consumption is the same as wild-type flies.

This is a valid concern. We have now included a brief discussion and a new result that addressed these possibilities. Below we offered our arguments against the ideas that enhanced feeding was responsible for the sucrose-induced lack of hardness discrimination and that reduced sugar-sensing by TMC neurons was responsible for *tmc* mutants’ clear discrimination in the presence of sucrose. We have also emphasized in revised material and method that we did not starve the flies beforehand.

1) We have shown before that in our arenas, flies’ interest to feed on a given substrate is generally *uncoupled* from their interest to lay eggs on that substrate (Yang, He and Stern, 2015; Yang et al.,2008). For example, when given a plain 1% agarose and a 100 mM sucrose-containing 1% agarose to explore in our arenas, flies preferred to feed on the sucrose-containing one but preferred to lay eggs on the plain 1% agarose (this result was also shown in Figure 4—figure supplement 1D). Similarly, when given a bitter+sucrose 1% agarose and a sucrose-only 1% agarose to explore, they preferred to feed on the sucrose-only substrate but to lay eggs on the bitter+sucrose substrate. (We note that Joseph et al., 2012 reported a similar observation where they found that when given a molasses food vs. a lobeline-containing molasses food to choose from, flies preferred to lay eggs on the bitter, lobeline-containing molasses food).

2) We have conducted a feeding preference experiment suggested by the reviewer (using the same arenas we used for egg-laying). We found that neither WT nor *tmc* mutants showed a feeding preference for the 0.5% agarose when given a 100 mM sucrose-containing 0.5% agarose and a 100mM sucrose-containing 1.5% agarose substrates to choose from (see new Figure 4—figure supplement 3). Thus, while WT and *tmc* mutants both showed lack of feeding discrimination when choosing between two sucrose-containing substrates of different hardness, they behaved very differently when choosing between the same two substrates for egg-laying – *tmc* mutants strongly favored the softer one for egg-laying but WT did not. This result again suggests that feeding and egg-laying preferences are not coupled in our arenas.

3) Two lines of evidence suggest that *tmc* mutants did not have impaired ability to sense sucrose. First, we have shown that *tmc* mutants and WT showed comparable discrimination of substrates that contained different levels of sucrose but were made with same concentration of agarose (see Figure 4—figure supplement 1D); second, Zhang et al., 2016 have shown by e-phys recording that MD-L neurons from WT and *tmc* mutants exhibited comparable levels of responses to sucrose.

4) Jeong et al., (2016) showed that activation of labellar mechanosensory neurons expressing Nanchung suppresses sugar neurons by means of axo-axonic inhibitory transmission. In this context, one would expect that sugar neuron transmission is dampened by harder substrates (e.g. 1.5% agarose), which could in principle suppress the synaptic output of sugar neurons. In this context, flies are expected to show preference for softer substrate. Similarly, Zhang et al., (2016) showed that the optogenetic activation of TMC neurons leads to a decrease of sucrose induced PER in flies, implying that the mechanosensory neurons are in fact inhibiting the sugar sensory neurons, not the other way around. However here, the authors show that the sugar neurons are in fact activating TMC neurons. The authors need to perform additional experiments to address this discrepancy or at least provide some explanation in the Discussion section.

This is a valid point that we should have discussed. Here we offered our explanations and we have included a discussion on this point in our revised manuscript. We believed findings reported in Jeong et al., (2016) and Zhang et al., (2016) are not incompatible with ours.

1) We showed that sweet neurons can act to potentiate the output of TMC neurons through presynaptic facilitation, these two reports showed that NOMPCneurons – and potentially also TMCneurons – can act to inhibit the output of the sweet neurons through presynaptic inhibition. These two forms of modulations can co-exist.

2) The reciprocal nature of the interactions between sweet neurons and the mechanosensitive neurons indeed raised the concern that 100 mM sucrose may exert less facilitation on the output of TMCneurons when animals were sampling 1.5% agarose than when animals were sampling 0.5% agarose (as output of the sweet neurons were expected to be dampened more on the harder 1.5% substrate). Consequently, this is expected to cause TMC neurons to experience stronger sucrose-induced facilitation on the softer substrate than on the harder one, thus narrowing the perceived the differences in hardness of 0.5% vs. 1.5% agarose (mediated by TMC neurons). Therefore, the modulation reported by Zhang et al., and Jeong et al., would not produce outcomes that contradict ours.

3) We suspect the impact of substrate hardness on output of sweet neurons might be less significant on the decisions we described in this manuscript. Jeong et al., found that different concentrations (0.2% – 2%) of agarose can impact behavioral preference for 0.5 mM vs. 1.0 mM sucrose. The sucrose we used was 100 mM. Further, while Jeong et al., and Zhang et al., both showed that activation of mechanosensitive neurons can inhibit PER – a direct behavior readout for activation of sweet neurons, it is unclear whether 0.5% vs. 1.5% agarose may exert significantly different levels of inhibition on PER. For example, in Zhang et al., strong PER suppression was observed by comparing the impact of 3% agarose vs. 0% agarose.

5) Schwartz et al., (2012) showed that the positive or negative influence of sugar on oviposition depends on the size of the egg laying chamber. The sugar-induced oviposition avoidance observed by the authors is likely an artifact of the small chamber size. It is unclear whether the interesting observation they made in this study is also influenced by the chamber size. The authors should at least conduct some of their key experiments in larger egg-laying chambers to see whether their conclusion still holds.

This is another very valid point. To address this question, we have now examined how WT and *tmc* mutants discriminated 0.5% vs. 1.5% agarose in the presence and absence of 100 mM sucrose in arenas that are ~ 8X the size of our regular arenas. We found that flies did not behave significantly differently when choosing substrates of different hardness in regular vs. large arenas: (1) both WT and *tmc* mutants preferred the 0.5% over 1.5% agarose in the large arenas, (2) 100 mM sucrose caused WT but not *tmc* mutants to reduce preference for 0.5% agarose in the large arenas. This result is now shown in new Figure 4—figure supplement 2B and C.